# Rapid proliferation due to better metabolic adaptation results in full virulence of a filament-deficient *Candida albicans* strain

Christine Dunker [1], Melanie Polke[1,5], Bianca Schulze-Richter [1,6], Katja Schubert[1], Sven Rudolphi [1], A. Elisabeth Gressler [1,6], Tony Pawlik [1], Juan P. Prada Salcedo[2], M. Joanna Niemiec [1], Silvia Slesiona-Künzel[1], Marc Swidergall [3], Ronny Martin [4], Thomas Dandekar [2] & Ilse D. Jacobsen [1]✉

The ability of the fungal pathogen *Candida albicans* to undergo a yeast-to-hypha transition is believed to be a key virulence factor, as filaments mediate tissue damage. Here, we show that virulence is not necessarily reduced in filament-deficient strains, and the results depend on the infection model used. We generate a filament-deficient strain by deletion or repression of *EED1* (known to be required for maintenance of hyphal growth). Consistent with previous studies, the strain is attenuated in damaging epithelial cells and macrophages in vitro and in a mouse model of intraperitoneal infection. However, in a mouse model of systemic infection, the strain is as virulent as the wild type when mice are challenged with intermediate infectious doses, and even more virulent when using low infectious doses. Retained virulence is associated with rapid yeast proliferation, likely the result of metabolic adaptation and improved fitness, leading to high organ fungal loads. Analyses of cytokine responses in vitro and in vivo, as well as systemic infections in immunosuppressed mice, suggest that differences in immunopathology contribute to some extent to retained virulence of the filament-deficient mutant. Our findings challenge the long-standing hypothesis that hyphae are essential for pathogenesis of systemic candidiasis by *C. albicans*.

[1] Research Group Microbial Immunology, Leibniz Institute for Natural Product Research and Infection Biology – Hans Knoell Institute, Beutenbergstraße 11a, Jena, Germany. [2] Department of Bioinformatics, Biocenter, Am Hubland, University of Würzburg, Würzburg, Germany. [3] The Lundquist Institute for Biomedical Innovation at Harbor UCLA Medical Center, David Geffen School of Medicine at UCLA, Los Angeles, CA, USA. [4] Institute for Hygiene and Microbiology, University of Würzburg, Würzburg, Germany. [5] Present address: Laboratory Dr. Wisplinghoff, Department of Molecular Biology, Horbeller Strasse 18-20, Cologne, Germany. [6] Present address: Institute of Immunology, Molecular Pathogenesis, Center for Biotechnology and Biomedicine (BBZ), College of Veterinary Medicine, Leipzig University, Deutscher Platz 5, Leipzig, Germany. ✉email: ilse.jacobsen@hki-jena.de

The polymorphic yeast *Candida albicans* colonizes up to 70% of the human population as a commensal on mucosal surfaces[1,2]. As an opportunistic fungal pathogen, however, *C. albicans* is able to cause superficial as well as life-threatening systemic infections promoted by disturbances in the microbiota and impaired host defenses[3,4]. Despite antifungal therapy, disseminated candidiasis is associated with high mortality rates of up to 50%[5,6].

The reversible transition of spherical budding yeast to pseudohyphal or hyphal filaments is promoted in vivo by body temperature, serum, physiological pH and elevated $CO_2$ concentration[7,8]. Nevertheless, both morphologies are present in tissues during systemic infection[9]. While filaments facilitate tissue invasion, damage and escape from host cells[10,11], yeast cells are believed to be important for mucosal colonization, dissemination through the bloodstream, adherence to endothelial cells and biofilm formation[12]. Furthermore, yeast and hypha are recognized differentially by immune cells[13]. Saville et al. dissected the relative contribution of *C. albicans* filamentation to virulence during systemic infection by using a regulable expression system, placing one copy of the negative regulator of filamentation *NRG1* under the control of a tetracycline-regulable promotor[14]. Mice challenged intravenously with the tet-*NRG1* strain succumbed to infection when hyphal growth was permitted but survived when fungal cells were enforced to grow in the yeast form. Similarly, other *C. albicans* yeast-locked mutants, such as the *cph1Δ/Δ efg1Δ/Δ* double and *hgc1Δ/Δ* mutant have been shown to be avirulent or are strongly attenuated in virulence[15,16]. On the other hand, mutants locked in the filamentous form like *tup1Δ/Δ* or *nrg1Δ/Δ* are less virulent as well[17–19], implying that morphological plasticity is essential for virulence in murine disseminated candidiasis.

One of the factors required for maintenance of hyphal growth, invasion and damage of *C. albicans* is encoded by *EED1* (Epithelial Escape and Dissemination 1[7,20]). A homozygous *eed1Δ/Δ* deletion mutant is still able to initiate germ tube formation, but fails to elongate these into hypha and eventually switches back to yeast cell growth[7,20,21]. Here we show that the *eed1Δ/Δ* mutant is fully virulent in a murine model of systemic candidiasis and confirmed this finding using a tetracycline-regulable expression system to induce or repress *EED1* in *C. albicans* in vivo. Virulence of these filament-deficient mutants is associated with high yeast proliferation rates in vivo and enhanced growth in vitro on nutrient sources likely encountered in the host, suggesting metabolic adaptation as the underlying mechanism for retained virulence in the absence of filamentation. Analysis of host responses furthermore supports a contribution of altered immunopathology to *C. albicans* virulence in the absence of filamentation.

## Results

### *EED1* is required for virulence in a murine intraperitoneal infection model.
Hypha formation has been shown to be essential for tissue invasion and damage in vitro and in vivo[11,14,22]. Consistent with this, the *eed1Δ/Δ* mutant, unable to maintain hyphal growth, was not able to damage renal, hepatic and oral epithelial cells in vitro within 24 h of co-incubation (Fig. 1a). To assess the impact of *EED1* for tissue invasion and damage in vivo, we employed an intraperitoneal infection model. In this model, *C. albicans* filamentation facilitates invasion from the peritoneal cavity into intraperitoneal organs such as liver, spleen and pancreas[23]. In line with the in vitro results, deletion of *EED1* led to reduced damage of liver and pancreatic tissue indicated by lower serum levels of tissue-specific enzymes and lower clinical scores compared to mice infected with the wild type strain

(WT; Supplementary Fig. 1). This attenuated virulence phenotype was confirmed using a conditional knock out mutant (t-EED1), in which one *EED1* allele was deleted and the other was placed under the control of a tet-OFF promoter[14]. In the presence of doxycycline (t-EED1 + ), repressed *EED1* expression leads to yeast cell growth. In absence of doxycycline (t-EED1−), the gene is constitutively expressed resulting in increased filamentation on solid media (Supplementary Fig. 2a) and filamentation comparable to the WT in liquid media (Supplementary Fig. 2b). In the presence of doxycycline, mice developed significantly less clinical symptoms after intraperitoneal infection with the t-EED1+ yeast compared to the WT and the t-EED1− filamentous strain 24 h post infection (p.i.) (Fig. 1b). Histological analysis showed t-EED1+ yeast that remained in the upper tissue layers below the liver capsule whereas WT and t-EED1− hyphae invaded deeply into liver parenchyma (Fig. 1c). Furthermore, infection with WT and t-EED1− induced significantly increased serum levels of alanine aminotransferase (ALT) and pancreatic amylase indicating liver and pancreatic injury, respectively (Fig. 1d). In contrast, reduced tissue invasion of the t-EED1+ yeast resulted in serum enzyme levels of ALT and pancreatic amylase comparable to the uninfected control. Interestingly, a significantly higher fungal burden was recovered from liver tissue of mice infected with either the t-EED1+ or *eed1Δ/Δ* mutant compared to the respective WT (Fig. 1e and Supplementary Fig. 1c). Thus, tissue damage and clinical symptoms correlated with the ability to form hyphae in the intraperitoneal infection model, and deletion or repression of *EED1* resulted in attenuation of virulence despite a higher liver fungal burden.

### Deletion of *EED1* affects dissemination from the gastrointestinal tract in immunosuppressed mice.
It is generally believed that the gastrointestinal tract is the main reservoir for *C. albicans* in humans[24,25]. In order to disseminate systemically, the fungus needs to translocate across the mucosal barrier[24,25]. Consistent with the in vitro results for renal, hepatic and oral epithelial cells, deletion of *EED1* significantly reduced the ability of *C. albicans* to damage enterocytes under normoxic and hypoxic conditions (Fig. 2a), resulting in maintained barrier function and reduced translocation in vitro (Fig. 2b, c). Colonization of the intestinal tract was moderately increased for *eed1Δ/Δ* compared to the WT in antibiotic-treated immunocompetent mice, but not in mice treated with cyclophosphamide (Fig. 2d, e). Dissemination of the *eed1Δ/Δ* mutant to internal organs was, however, only observed upon cyclophosphamide treatment (Fig. 2f), where a significantly higher number of colony forming units (CFU) in kidneys were observed for *eed1Δ/Δ* compared to the WT. Of note, while cyclophosphamide reduces the numbers of immune cells and also impacts the intestinal barrier function[26,27], we did not observe any symptoms of disseminated disease in the mice. This suggests that the treatment was not sufficient to induce the level of immunosuppression and/or intestinal damage required for translocation of a sufficiently high number of fungal cells to cause disease[27]. Nonetheless, these results support the concept that growth in the yeast form is beneficial for intestinal colonization[28,29] (at least in immunocompetent mice) and dissemination[12], but that filamentation is required for active translocation[22].

### *EED1* is not essential for virulence in a systemic infection model.
Based on the in vitro data and results from the intraperitoneal infection model, we expected the *EED1* mutants also to be attenuated in a murine systemic (intravenous) infection model that mimics catheter-associated disseminated candidiasis in humans. Surprisingly, the *eed1Δ/Δ* mutant showed enhanced

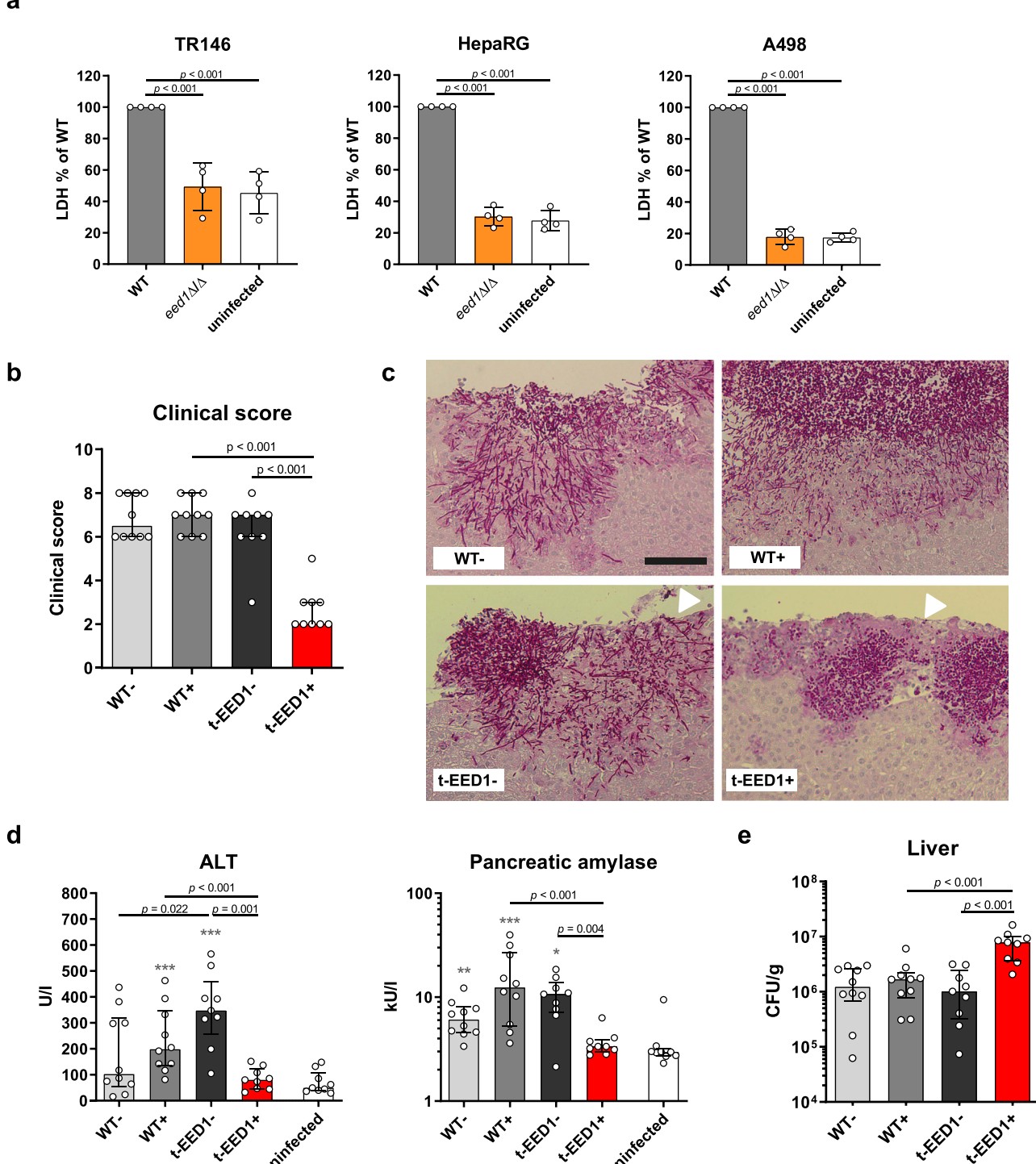

virulence compared to the WT when low infectious doses were used (Fig. 3a) whereas with a high infectious dose mortality was delayed for t-EED1 + yeast (Fig. 3b). Intermediate infectious doses led to virulence and disease progression indistinguishable to the WT when t-EED1 was either forced to grow as yeast or as hyphae (Fig. 3c, Supplementary Fig. 3). Histology confirmed that t-EED1 + grew as yeast in vivo (Fig. 3d). Consistent with previous reports, experiments conducted in parallel with the tet-NRG1 strain[14] resulted in clinical disease only when filamentation was facilitated by the presence of doxycycline (Supplementary Fig. 4a-c[14]). We also noted that mice infected with the WT (THE1-CIp10) reached humane endpoints earlier when receiving

doxycycline; this is likely a consequence of lower water uptake in these groups (see method section for details). Determination of CFU in mice infected with the intermediate dose showed similar initial fungal burden 6 h p.i. in kidney, liver, brain and spleen (Fig. 3e). However, already 24 h p.i. we observed significantly increased (~25-fold higher) fungal burden in the kidneys of mice infected with t-EED1 + yeast compared to the filamentous strains. Likewise, fungal burden increased for tet-NRG1- yeast, reaching significantly higher levels 72 h p.i. compared to the WT-, but in comparison to t-EED1 + yeast declined until the end of the experiment (Supplementary Fig. 4d). Furthermore, fungal burden of t-EED1 + yeast in the brain continued to increase until mice

**Fig. 1 Hyphal elongation of *C. albicans* is required for tissue damage in vitro and virulence in an intraperitoneal infection model. a** Damage of oral (TR146), hepatic (HepaRG) and renal (A498) epithelial cells was quantified by measuring the release of lactate dehydrogenase (LDH) into the supernatant after co-incubation for 24 h with WT (SC5314) or *eed1Δ/Δ* mutant. Uninfected cells served as negative control. LDH released by WT was set to 100%. Data are presented as mean ± SD from four biologically independent experiments. Within each experiment, supernatants from three wells infected by the same strain were pooled and LDH was measured. Data was analyzed by two-tailed students *t*-test. **b–e** t-EED1 + yeasts are attenuated in invasion, damage and virulence potential despite higher organ fungal loads in the intraperitoneal infection model 24 h post infection. Mice were infected intraperitoneally with $1 \times 10^8$ cells of WT (THE1-Clp10) or t-EED1 in the presence (+) or absence (−) of doxycycline supplied via the drinking water. Data derived from two independent experiments, WT− and WT+ $n = 10$, t-EED1− and t-EED1+ $n = 9$ animals per group. **b** Repression of *EED1* by doxycycline led to significantly reduced clinical symptoms in mice. The semiquantitative clinical score was determined by assessing fur, coat and posture, behavior and lethargy, fibrin exudation and other symptoms like diarrhea. The score ranges from 0 (no symptoms) to 10 (severe illness). **c** Representative images of periodic acid-Schiff stained histological sections of liver tissue 24 h p.i., fungal cells are stained purple. Arrows point towards the liver surface (capsule, partially destroyed). The scale bar represents 100 μm and applies to all images. **d** Damage of liver and pancreas was quantified by measuring serum levels of alanine aminotransaminase (ALT) and pancreatic amylase, respectively. Enzyme levels of uninfected mice ($n = 9$) served as negative control. **e** Fungal burden in the liver. **b**, **d**, **e** Median and interquartile range are shown, two-sided Mann–Whitney test. Asterisks above bars represent significant differences compared to the uninfected control *$p \leq 0.05$; **$p \leq 0.01$; ***$p \leq 0.001$. Source data are provided as a Source Data file.

became moribund, whereas after infection with filamentous strains cell numbers peaked 24 h p.i. and were stable or declined thereafter. After an initial decline, the CFU of the t-EED1 + yeast stabilized and increased in liver and spleen while they continuously declined after infection with the WT, t-EED1- or tet-*NRG1* yeast or hypha (Fig. 3e; Supplementary Fig. 4d). To test the possibility that differences in fungal load were due to the underestimation of CFU plated from filamentous *C. albicans* strains, we performed quantitative PCR targeting the *C. albicans* 18 S rRNA gene *RDN18* in complex DNA isolated from infected kidney homogenates[30]. A strong correlation ($R^2 = 0.7076$) between CFU count and DNA content irrespective of the fungal morphology (Supplementary Fig. 5) supports the CFU results.

**During hematogenously disseminated candidiasis *EED1* deficiency leads to delayed renal cytokine production but increased immune cell infiltration at later time points.** We hypothesized that the increased fungal burden observed for t-EED1 + yeast could be the result of an altered early innate immune response leading to reduced fungal killing. Therefore, we quantified immune cells in the kidney over the course of infection. Leukocytes started to infiltrate kidneys 24 h p.i. and no difference was observed in the overall number or subpopulations of leukocytes 24 and 48 h p.i. between WT + and t-EED1 + yeast (Supplementary Figs. 6–7). However, morphology-dependent significant differences between WT + and t-EED1+ were observed 72 h p.i.: Total leukocyte numbers were approximately 2-fold higher in response to t-EED1+ yeast (Fig. 4a), with increased numbers of monocytes (4.5-fold), macrophages (3.2-fold) and DCs and NK cells (2-fold). This coincided with 190-fold higher yeast fungal burden compared to WT+ (Fig. 3e). Interestingly, significantly less leukocytes accumulated in kidneys after infection with the t-EED1− filamentous strain compared to WT− (Supplementary Fig. 8), despite comparable CFU 72 h p.i. (Fig. 3e). While renal cytokine levels induced by the filamentous t-EED1− were comparable to the WT− throughout the course of infection (Supplementary Figs. 9–10), less pro-inflammatory cytokines were induced by t-EED1+ yeast than by WT+ 24 h p.i. (Fig. 4b), despite higher fungal burden of t-EED1+ yeast. In contrast, cytokine levels were comparable 48 h p.i., and a tendency towards increased cytokine release, with significant higher amounts of IL-18, IP-10, RANTES and IFN-γ in response to t-EED1+ yeast, was observed 72 h p.i. (Fig. 4b). In moribund mice no morphology-dependent difference in cytokine levels was observed (Supplementary Fig. 10). To investigate whether the local differences in renal cytokine production were reflected by differences in systemic inflammation, we determined serum levels of the sepsis markers soluble triggering receptor expressed on myeloid cells

(sTREM-1) and neutrophil gelatinase-associated lipocalin (NGAL). NGAL and sTREM-1 levels increased rapidly and throughout the course of infection (Fig. 4c) irrespective of the fungal morphology.

Lower cytokine levels induced by t-EED1+ yeast in the presence of similar leukocyte numbers 24 h p.i. suggested differences in the interaction with leukocytes and/or epithelial cells and their interaction with fungal cells was therefore analyzed in vitro. Murine bone marrow (BM) neutrophils and BM-derived macrophages (BMDMs) were capable of phagocytosing and killing *C. albicans* WT and *eed1Δ/Δ* mutant to the same extent in vitro after 1 h and 2 h, respectively (Fig. 5a, b; Supplementary Fig. 11a). At that time point both strains had formed germ tubes (Supplementary Fig. 11b and Supplementary Fig. 12). After 6 h, the WT maintained filamentation while the *eed1Δ/Δ* mutant had switched back to yeast cell growth (Fig. 5c; Supplementary Fig. 12). Analysis at 6 h showed that a significantly larger proportion of *C. albicans eed1Δ/Δ* was killed by BMDMs compared to the WT. However, we noted that the relative increase in CFU of the *eed1Δ/Δ* mutant from 0 h to 6 h was considerably higher than for the WT both in the presence and absence of macrophages (Fig. 5d). This resulted in significantly higher numbers of *eed1Δ/Δ* than WT cells in the presence of BMDMs. In addition, the *eed1Δ/Δ* mutant showed a reduced capacity to damage BMDMs (Fig. 5e) and to stimulate TNF-α release by BMDMs 24 h p.i. (Fig. 5f). At that time wells that contained the *C. albicans* WT were completely covered by hyphae whereas wells with *eed1Δ/Δ* showed yeasts only and macrophages were visible from time to time (Supplementary Fig. 12). In contrast, activation of murine PMNs determined by the release of IL-10, IL-6 and TNF-α (Supplementary Fig. 11c) and production of reactive oxygen species (ROS; Supplementary Fig. 11d) was comparable between the strains. Infection of renal, hepatic and oral epithelial cells with the *eed1Δ/Δ* mutant induced no increase of pro-inflammatory cytokines compared to uninfected controls (Fig. 5g), likely due to the lack of damage caused by the mutant in this model (Fig. 1a). For oral epithelial cells, no significant increase was observed for the WT either, although a tendency of higher cytokine production was observed. Thus, the initial lower cytokine release in response to t-EED1+ yeast in vivo is likely due to reduced damage of and lower cytokine production by renal and hepatic cells and macrophages, which is balanced by the response of recruited immune cells at later time points.

**_eed1Δ/Δ_ yeast display enhanced growth on physiologically relevant carbon sources.** *EED1* depleted yeast were able to increase their cell number approximately 100-fold from 6 h to 24 h p.i. in murine kidneys (which equals a generation time of 2.6 h

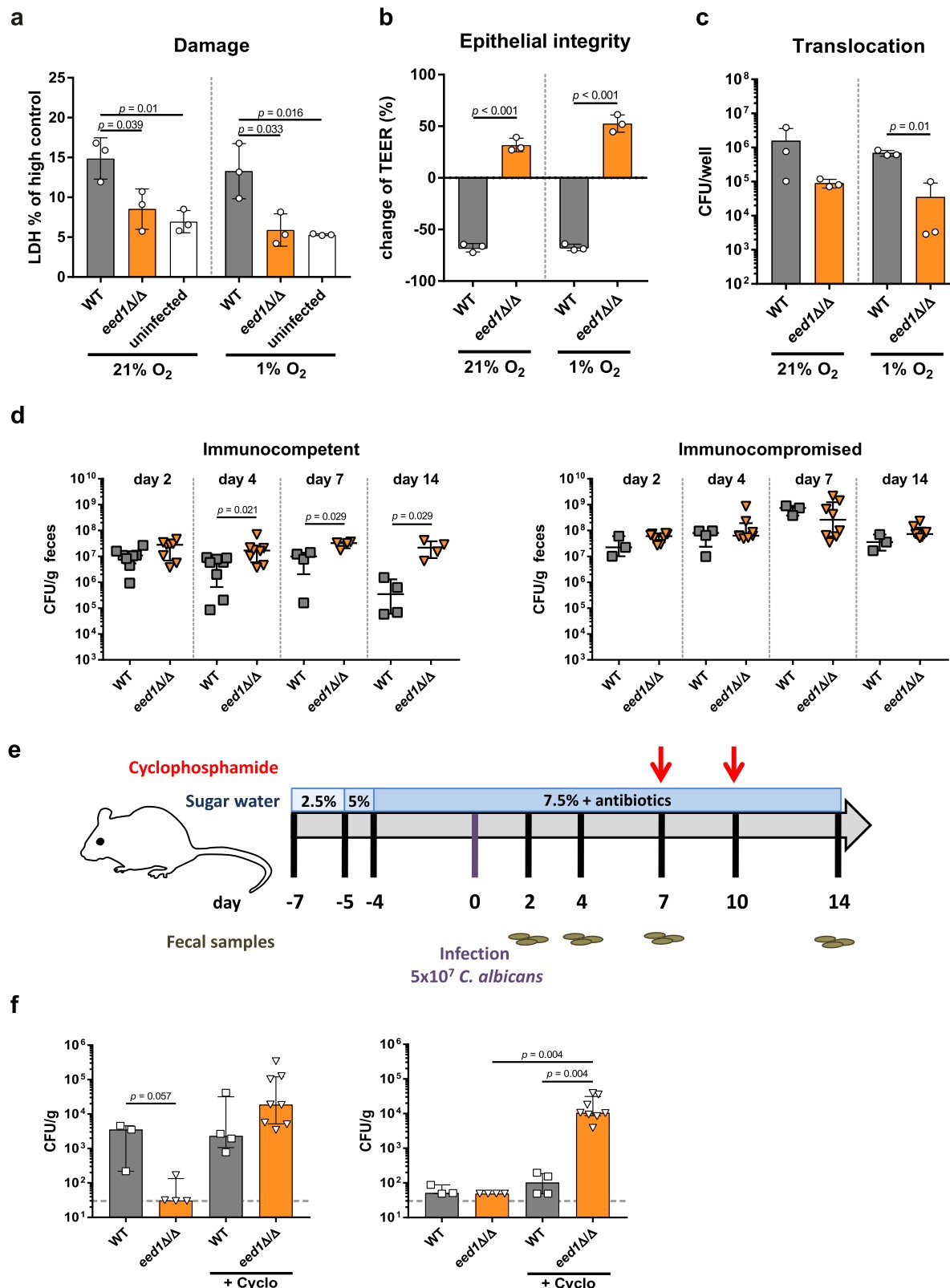

compared to 7.1 h for the WT). Since we found no evidence for reduced recognition or killing by immune cells, this observation cannot be explained by increased immune evasion resulting in less killing. We thus hypothesized that *EED1*-deficient yeasts are metabolically better adapted to acquire locally available nutrients resulting in increased proliferation. Therefore, we analyzed the growth of *C. albicans* WT and the *eed1*Δ/Δ mutant in the

presence of different fermentable and non-fermentable carbon sources at 37 °C (Fig. 6). Growth of both strains was similar with 2% glucose in complex (YPD) and defined media (SD). However, in the presence of physiological glucose concentrations of 0.1% in SD medium, the mutant reached lower cell densities at stationary phase. In YPD with 0.1% glucose, in contrast, the *eed1*Δ/Δ mutant continued to grow steadily after depletion of glucose

**Fig. 2 The *eed1Δ/Δ* mutant shows a reduced translocation capacity in vitro and in a gut dissemination model in immunocompetent but not in immunocompromised mice. a–c** C2BBe1 cells were infected with *C. albicans* WT (SC5314) or the *eed1Δ/Δ* mutant and incubated under normoxic (21% $O_2$) or hypoxic (1% $O_2$) conditions for 24 h. **a** Damage of C2BBe1 cells was quantified by measurement of LDH and is shown in % of high control (cells lysed with Triton X-100). Uninfected cells served as negative control. **b** Integrity of epithelial barrier was quantified by TEER measurement. Data are expressed as change of TEER 24 h post infection compared to TEER values prior to infection in percent. **c** Translocation across the epithelial barrier was assessed by CFU plating. **a–c** Graphs show the mean ± SD of three independent biological replicates, two-tailed students *t*-test. *p*-values are shown in the graph. **d** Fungal burden in feces of antibiotic-treated immunocompetent and mice rendered immunocompromised by cyclophosphamide treatment. Immunocompetent, day 2 and 4 $n = 8$; day 7 and 14 $n = 4$ mice per group, one experiment. Immunocompromised WT infected $n = 3$; *eed1Δ/Δ* $n = 8$, one experiment. **e** Schematic overview of the timeline for the gastrointestinal dissemination model. **f** Fungal burden in liver (left) and kidneys (right) of immunocompetent or cyclophosphamide-treated immunosuppressed (+Cyclo) mice colonized with either *C. albicans* WT or *eed1Δ/Δ* mutant 14 d post infection. WT $n = 3$, one experiment; WT + Cyclo and *eed1Δ/Δ* $n = 4$, one experiment; *eed1Δ/Δ* + Cyclo $n = 8$ mice, data from two independent experiments. Dashed lines indicate limit of detection. **d, f** Shown is the median with interquartile range, two-sided Mann–Whitney test, *p*-values are shown in the graph. Source data are provided as a Source Data file.

reaching higher ODs than the WT. Importantly, the *eed1Δ/Δ* mutant showed enhanced growth in the presence of the physiological relevant alternative carbon sources lactate, acetate, citrate, amino acids and the amino sugar N-acetyl-glucosamine (GlcNAc). Additionally, the *eed1Δ/Δ* mutant showed faster onset of growth in YCB-BSA compared to the WT indicating that this strain is highly proteolytically active. Furthermore, *eed1Δ/Δ* showed better growth than the WT in murine kidney homogenates. Taken together, these data suggest that the absence of *EED1* results in better metabolic adaptation in vivo, facilitating faster proliferation resulting in increased fungal burden.

To gain information on possible mechanisms mediating the enhanced proliferation of the *eed1Δ/Δ* mutant on alternative carbon sources, we performed RNAseq analysis of cells grown in SD with glucose at 30 °C for 10 h (designated as 0 h time point), followed by a shift to citrate or casamino acids as sole carbon source at 37 °C (2 h, 6 h, and 12 h; Supplementary Fig. 13a). Growth in SD with glucose resulted in yeast morphology for both strains (Supplementary Fig. 13b); on casamino acids, both strains formed germ tubes within 2 h, but only the WT formed hyphae at later time points (Supplementary Fig. 13b). With citrate as sole carbon source, filamentation was observed only for the WT (Supplementary Fig. 13b). In both, citrate and amino acid media, the *eed1Δ/Δ* mutant showed enhanced growth as observed by optical density as well as by determination of dry weight (Supplementary Fig. 14a). Principal component analysis (PCA) of the RNAseq data showed that biological replicates clustered together and that the transcriptome of both strains varied over time (Supplementary Fig. 14b). Comparison of *eed1Δ/Δ* mutant and WT for each condition and time point (with a log$_2$ fold change of 2.0 as cut off) revealed that only one yet uncharacterized gene (orf19.2962) was differentially expressed during yeast growth at the 0 h time point (2.1-fold down-regulated). After the shift to media containing either citrate or casamino acids as sole carbon sources, relatively few genes (<320) were differentially expressed in the mutant compared to the WT at any given time point, and the majority of differentially expressed genes (DEGs) were down-regulated in *eed1Δ/Δ* compared to the WT (Supplementary Fig. 15). Consistent with differences in morphology, down-regulated genes in both conditions included genes associated with filamentous growth, biofilm formation, and adhesion (e.g. *ECE1*, *HGC1*, *BRG1*, *UME6*, *HYR1*, *SAP5*, *HWP1* and *ALS1*; Supplementary Fig 16a). The expression of secreted aspartyl proteinases likewise coincided with morphology, with the hypha-associated genes *SAP4-6*[31] showing reduced transcription while transcription of the yeast-associated genes *SAP1* and *SAP3*[31] was enhanced in the *eed1Δ/Δ* mutant during growth on casamino acids (Supplementary Fig. 16b). The highest number of DEGs was observed at the 12 h time point (Fig. 7a), when the *eed1Δ/Δ* mutant grew as yeast but also the filamentous WT

switched back to yeast cell growth to some extent (Fig. 7b). Genes that were up-regulated in *eed1Δ/Δ* mutant compared to WT in both media included the ammonium permease *MEP1*, the putative 2-isopropylmalate synthase *LEU4*, the secreted yeast wall protein *YWP1*, the transcription factor *MSS11*, and 5 yet uncharacterized open reading frames. Most of the DEGs showed a medium-specific regulation. After 12 h of growth with citrate as sole carbon source Gene Ontology (GO) term analysis identified enrichment of genes associated with organic acid, (long-chain) fatty acid and small molecule biosynthetic and metabolic processes within the up-regulated DEGs in the *eed1Δ/Δ* mutant (Fig. 7c), which might explain the better growth of the mutant on citrate. Genes that were up-regulated exclusively in citrate included e.g., the dicarboxylic acid transporter *JEN2* and a key enzyme of gluconeogenesis, *PCK1*. Of note, the only significantly enriched GO term after 12 h of growth on casamino acids in the genes up-regulated in *eed1Δ/Δ* was found to be carbohydrate transport. However, individual genes that were significantly up-regulated included the broad specificity amino acid permease *GAP2*, the predicted amino acid transmembrane transporter *UGA5*, the GATA-type transcription factor *GAT1* that regulates nitrogen utilization, and the glyoxylate cycle enzyme isocitrate lyase *ICL1* (Supplementary Data 1), which might contribute to enhanced growth. In addition to *GAT1*, three other transcription factors (TFs) were up-regulated and 13 down-regulated in the *eed1Δ/Δ* mutant 12 h after shift to either citrate of casamino acids (Supplementary Fig. 16c). Several down-regulated TFs were associated with filamentous growth and thus, differential expression likely reflects differences in morphology (e.g., *UME6*, *SFL2*, *BRG1*, *TEC1*, *OFI1*[32], *ACE2*[33]). Counterintuitively, though, transcription of *MSS11*, encoding a factor interacting with Flo8 to activate transcription of hypha-specific genes[34], was higher in the mutant in both citrate and amino acid media. Similarly, expression of *RFX2* was reduced in the *eed1Δ/Δ* mutant – deletion of this gene, however, results in hyper-filamentation and increased expression of hypha-associated genes[35]. Interestingly, both *WOR1* and *WOR3* were significantly down-regulated after prolonged growth in both media, possibly indicating increased commitment of the *eed1Δ/Δ* mutant to grow as white cells. White cells have been shown to have a metabolic advantage over opaque cells with various nutrients at 37 °C[36], lead to higher renal fungal burden, and are more virulent in systemic infection models[37,38].

**High numbers of yeast cells result in renal damage in vivo**. To determine if immune cell recruitment together with high fungal burden during t-EED1 + infection was sufficient to cause renal damage in the absence of invading hyphae, we measured KIM-1, a biomarker for renal proximal tubule injury in the urine of infected mice[39,40]. KIM-1 levels increased after infection with all strains without significant differences (Fig. 8a). Additionally, we

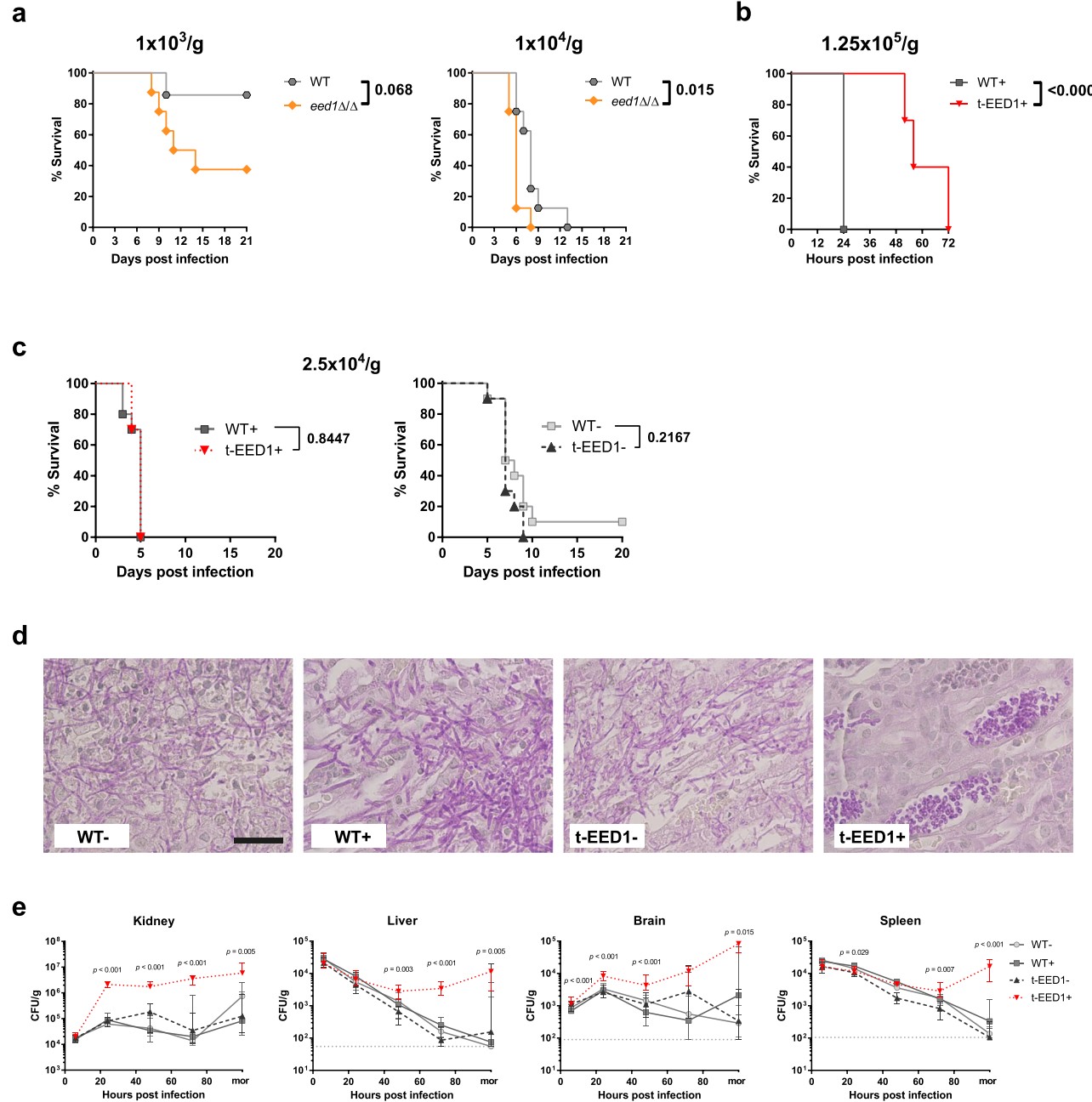

**Fig. 3 *C. albicans* retains its virulence potential in a mouse model of systemic candidiasis even in the absence of hyphal elongation due to repression or deletion of *EED1* accompanied by higher organ fungal loads.** Mice were intravenously infected with the following doses: **a** Low doses of $1 \times 10^3$ or $1 \times 10^4$ CFU/g body weight of WT (SC5314) or *eed1Δ/Δ* mutant. One experiment, $n = 8$ mice per group (except for WT $1 \times 10^3$, $n = 7$). **b** A high dose of $1.25 \times 10^5$ CFU/g body weight of WT (THE1-CIp10) or t-EED1 in the presence of doxycycline. One experiment, $n = 10$ mice per group. **c** An intermediate dose of $2.5 \times 10^4$ CFU/g body weight with WT (THE1-CIp10) or t-EED1 in the presence (+) or absence (−) of doxycycline. One experiment, $n = 10$ mice per group. **a–c** Survival was monitored over a course of 20 or 21 days and is shown as Kaplan–Meyer curve. Survival curves were compared using the two-sided Log-rank (Mantel–Cox) test, p-values are shown in the graph. **d** Representative images of PAS stained histological cross sections of kidneys from moribund mice infected with $2.5 \times 10^4$ CFU/g body weight with WT (THE1-CIp10) or t-EED1 in the presence (+) or absence (−) of doxycycline. Fungal cells are stained purple. Scale bar represents 20 μm and applies to all images. **e** Organ fungal loads of mice infected with $2.5 \times 10^4$ CFU/g body weight with WT (THE1-CIp10) or t-EED1 in the presence (+) or absence (−) of doxycycline. Shown is the number of CFU per gram organ from mice sacrificed 6, 24, 48 and 72 h post infection and from moribund mice (mor). Data are shown as median with interquartile range. Dashed gray lines indicate limit of detection. Moribund: one experiment, $n = 10$; Kidney: two independent experiments $n = 10$. Liver, spleen, brain: three independent experiments, $n = 15$. Two-sided Mann–Whitney test, asterisks indicate significant changes of t-EED1+ compared to WT+ groups at the indicated time points. *$p \leq 0.05$; **$p \leq 0.01$; ***$p \leq 0.001$.

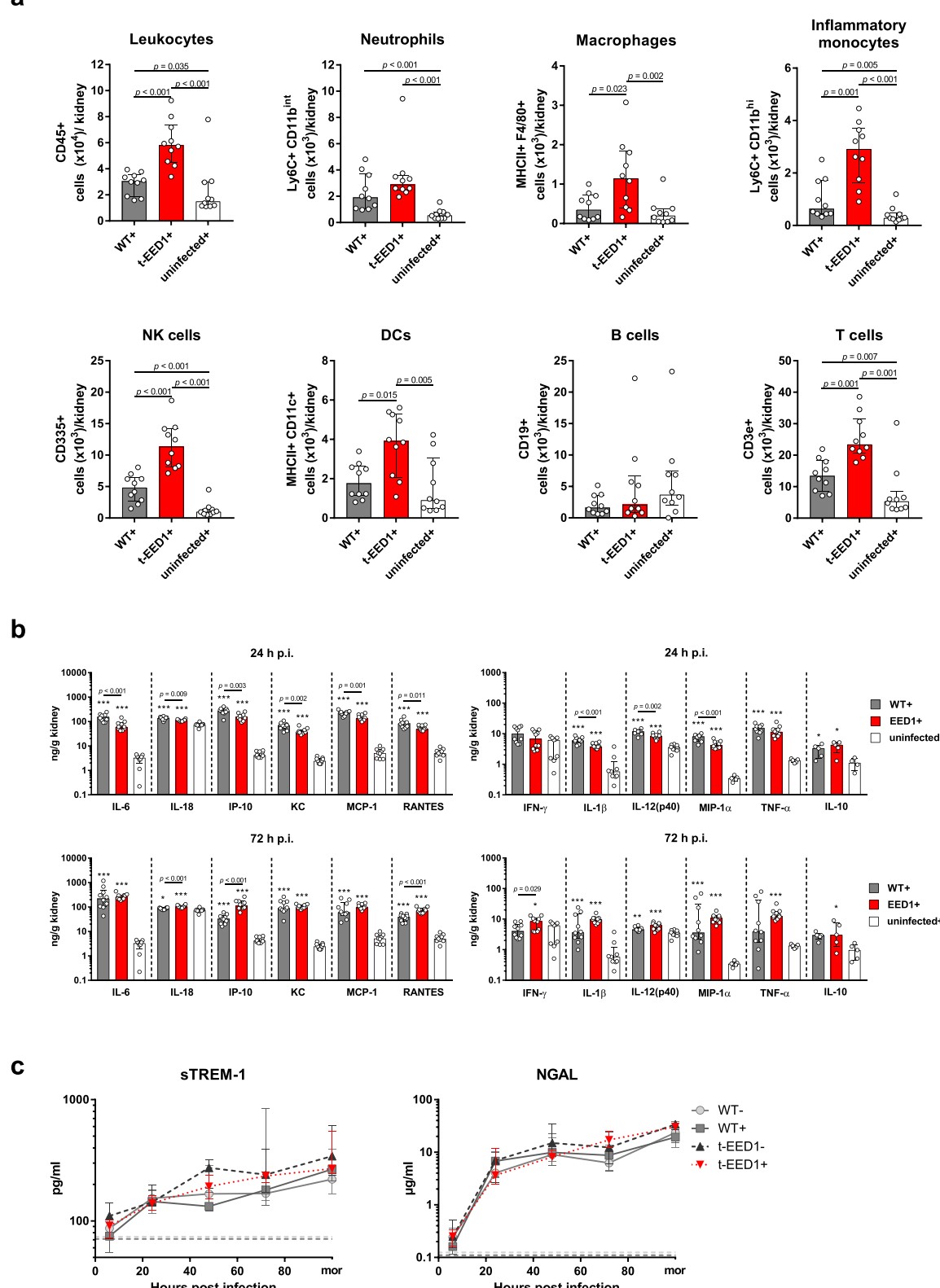

evaluated kidney function by measuring serum creatinine and blood urea nitrogen (BUN) levels. Both increased at comparable rates, reaching highly elevated levels indicating severe renal dysfunction[41,42] in moribund mice only (Fig. 8b, c). Finally, the same extent of apoptosis was observed in the kidneys of moribund mice infected with either WT or t-EED1 (Fig. 8d, e). Therefore, enhanced fungal proliferation can compensate for the reduced capacity of hypha-mediated damage in *C. albicans* lacking expression of *EED1*.

**Rapid yeast proliferation in combination with immunopathology contributes to mortality.** To investigate whether mortality caused by the yeast form is the result of high organ fungal loads interfering with organ function or whether

**Fig. 4 Increased immune cell infiltration in kidneys of mice 72 h after systemic infection with t-EED1+ yeast coincides with increased local cytokine production while systemic inflammation is not affected.** Mice were systemically infected with an intermediate dose of $2.5 \times 10^4$ CFU/g body weight of WT (THE1-CIp10), t-EED1 or remained uninfected in the presence (+) of doxycycline. **a** Immune cells infiltrating the kidney 72 h post infection. Shown are absolute numbers of living immune cells per kidney. Two independent experiments, $n = 10$. **b** Cytokine levels were measured in kidney homogenates 24 h and 72 h post infection. Asterisks above bars represent significant differences compared to the uninfected control. Two independent experiments, $n = 10$, except for uninfected+, $n = 9$. MIP-1α, TNF-α for uninfected+, $n = 5$. IL-10 data derived from one experiment, $n = 5$. **c** Markers for systemic inflammation sTREM-1 and NGAL were quantified by ELISA in serum of mice after 6, 24, 48, 72 h and when mice become moribund (mor). Dashed lines indicate median serum protein levels of uninfected controls in the presence (dark gray) or absence of doxycycline (light gray). Two independent experiments. sTREM-1 $n = 10$, except for 6 h $n = 6$ and mor $n = 5$; for WT- 24 h $n = 8$ and 48 h $n = 9$. NGAL $n = 10$, except for WT− 24 h $n = 9$, t-EED1− 24 h $n = 8$ and 72 h $n = 9$. **a–c** Shown is the median and interquartile range, two-sided Mann–Whitney test. **a**, **b** $p$-values are shown in the graph and **b** asterisks indicate significant differences in comparison to the uninfected control (*$p \leq 0.05$; **$p \leq 0.01$; ***$p \leq 0.001$). Source data are provided as a Source Data file.

immunopathology is the driving force in the pathogenic process, mice successfully depleted of peripheral neutrophils and monocytes (Supplementary Fig. 17) were intravenously infected with low infectious doses ($1 \times 10^2$ and $1 \times 10^3$ CFU/g body weight). Although the *eed1Δ/Δ* mutant led to 100% mortality in immunosuppressed mice, this was significantly delayed in comparison to WT strain infections (Fig. 9a). However, in immunocompetent mice challenged with $1 \times 10^3$ CFU/g body weight, the obverse effect was observed: while 6/7 mice survived in the WT, only 3/8 mice survived in the *eed1Δ/Δ* group until the end of the experiment (Fig. 3a). In addition, histological analysis of renal tissue showed more pronounced immune cell infiltrations in response to *eed1Δ/Δ* yeast in immunocompetent mice (Fig. 9b), suggesting some contribution of immunopathology to yeast driven mortality. In the absence of neutrophils and monocytes, fungal burden of WT and *eed1Δ/Δ* mutant progressively increased during the course of infection. However, organ fungal load of the *eed1Δ/Δ* mutant exceeded that of the WT in all organs tested, with significant differences detectable as early as 12 h post infections in liver and kidneys (Fig. 9c). At the humane endpoint, the fungal burden of the *eed1Δ/Δ* mutant exceeded those of the WT in all organs tested (Fig. 9c and Supplementary Fig. 18a) with a 150-fold and 280-fold increase in kidney and spleen fungal burden, respectively. Furthermore, the urinary KIM-1 to creatinine ratio steadily increased after infection with *C. albicans eed1Δ/Δ*, indicating that filaments are not essential for induction of renal injury (Fig. 9d). However, kidney function as determined by measurement of BUN in serum of immunosuppressed mice was not affected at any time after infection with WT or *eed1Δ/Δ* mutant (Supplementary Fig. 18b), possibly because the time to the humane endpoint was too short to allow accumulation of waste products by impaired renal clearance.

## Discussion

*C. albicans* hypha formation has been linked to invasion and damage, and *C. albicans* strains locked in either the yeast or hyphal morphology were repeatedly found to be less virulent in systemic infection models[15–18]. Consequently, morphological plasticity is considered as an important virulence trait of *C. albicans*[43,44]. Consistent with the role of filamentation for invasion and damage, repression or absence of *EED1* in *C. albicans*, interfering with hyphal elongation and leading to yeast growth, resulted in significantly reduced capacity to damage epithelial cells in vitro, and reduced virulence in a murine intraperitoneal infection model in vivo although the mutant was present in higher numbers than the WT in the liver. The observation that following intraperitoneal injection *EED1*-deficient cells superficially invaded the liver parenchyma but remained below the liver capsule, without causing detectable damage, resembles the described behavior of the *eed1Δ/Δ* mutant in a reconstituted human epithelium model[20]. Thus, our results support previous research demonstrating that hyphae are important for tissue

infiltration. In the gut, however, *C. albicans* can be found in both yeast and hyphal form[28,45,46]. Some studies found higher colonization rates for strains locked in the yeast morphology (e.g., the *cph1Δ/Δ efg1Δ/Δ* double mutant and the *hgc1Δ/Δ* mutant[28]), whereas enforced filamentous growth resulted in lower intestinal colonization (e.g. *C. albicans nrg1Δ/Δ* and *tup1Δ/Δ* mutants[27,28], repression of *TUP1*[29], and constitutive expression of *UME6*[28,45]). We likewise observed higher colonization rates for the *eed1Δ/Δ* mutant in immunocompetent mice, supporting the concept that yeast growth favors intestinal colonization. How *C. albicans* is translocating across the intestinal barrier remains unknown. In vitro, translocation requires filamentation[22], but low level translocation has been observed in immunocompetent mice independent of fungal morphology[28]. This has been suggested to be mediated by transport via host cells that sample the intestinal content[28]. We observed reduced basal dissemination of the *eed1Δ/Δ* mutant to the liver, which might be a consequence of altered interaction of the mutant with dendritic cells or M cells mediating entry into the bloodstream by lumen sampling or transcytosis[28,47] in the absence of hyphal formation, but this aspect was not investigated further in this study. In contrast, increased fungal burden in liver and kidney was observed in *C. albicans eed1Δ/Δ* colonized mice treated with cyclophosphamide, and in these mice kidney CFU were significantly higher for the *eed1Δ/Δ* compared to the WT. However, this data does not allow any conclusion on the translocation process, as the higher fungal load could also be a consequence of increased proliferation of the mutant within this organ. It should furthermore be noted that the fungal burden in kidneys following translocation from the gut was substantially lower than that observed after intravenous infection, which explains why colonized mice did not develop signs of systemic candidiasis upon cyclophosphamide treatment.

Surprisingly, the filamentation defect caused by *EED1* deficiency did not impair virulence in a systemic infection model when intermediate infectious doses were injected directly into the bloodstream. Remarkably, infectious doses lower than $10^4$ cells per gram body weight resulted in increased virulence of the *eed1Δ/Δ* mutant compared to the WT, whereas with high infectious doses or in absence of neutrophils and monocytes mortality was delayed but still reached 100%. In the murine model of hematogenously disseminated candidiasis, infection is established via the lateral tail vein mimicking disseminated *C. albicans* infections in humans[48]. While virtually all organs can get affected during infection the kidney is not able to control fungal growth[49]. The fungus reaches the kidneys via the renal artery and the afferent arterioles. In order to invade into tubules and the renal parenchyma, the fungus needs to adhere to endothelial cells and pass through into the renal cortex. This step probably does not require fungal invasion by hypha formation, as tet-*NRG1*- yeast and a *hgc1Δ/Δ* mutant defective in hyphal formation were shown to traverse from the bloodstream to the renal endothelium[14,16]. Although attenuated in its damage capacity, the *eed1Δ/Δ* mutant

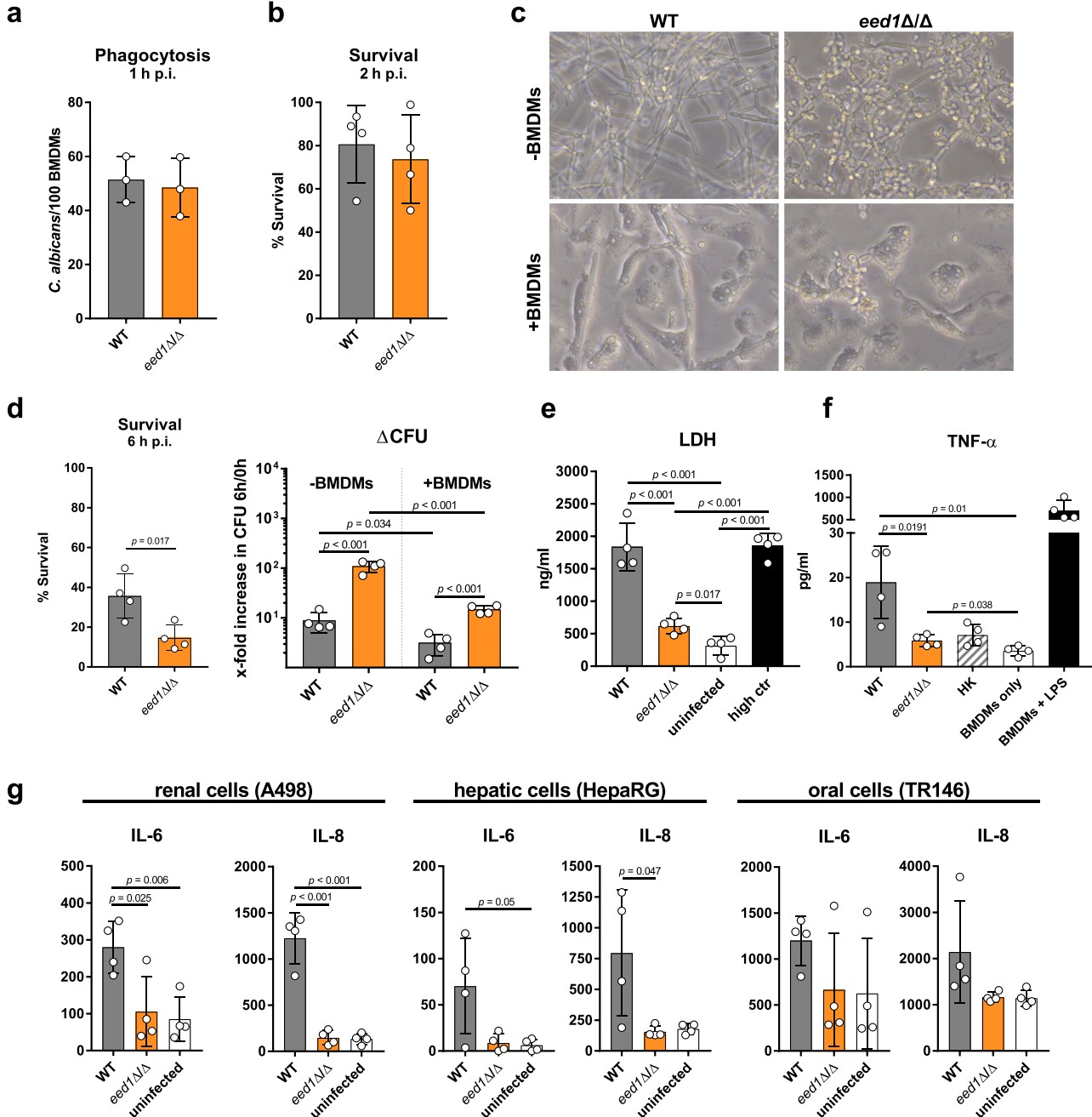

**Fig. 5 *eed1Δ/Δ* yeast are less efficient in damaging and inducing cytokine responses of bone marrow-derived macrophages (BMDMs) and epithelial cells.** Murine bone marrow-derived macrophages (BMDMs) were infected at an MOI of 1 with *C. albicans* WT (SC5314) or *eed1Δ/Δ* mutant and incubated at 37 °C and 5% CO$_2$. **a** Phagocytosis calculated as phagocytic index (the number of *C. albicans* cells phagocytosed by 100 BMDMs within 1 h of incubation). Mean ±SD from three biologically independent experiments. **b** Fungal survival was analyzed after co-incubation with BMDMs for 2 h by CFU plating. Survival was normalized to *C. albicans* controls incubated in the absence of immune cells. **c** Fungal morphology of WT and *eed1Δ/Δ* mutant in the absence (−) or presence (+) of BMDMs after 6 h of incubation. Scale bar represents 20 μm and applies to all images. **d** Fungal survival was analyzed after co-incubation with BMDMs for 6 h by CFU plating. Left: Survival was normalized to *C. albicans* controls incubated in the absence of immune cells. Right: To account for the observation that the mutant replicates faster than the WT in the absence of macrophages, data is shown as fold increase in CFU from 0 h to 6 h with and without BMDMs. **e** Damage of BMDMs was quantified by measuring lactate dehydrogenase (LDH) release into the supernatant after 24 h of co-incubation with *C. albicans* WT or *eed1Δ/Δ* mutant. Uninfected cells served as negative control and Triton X-100 lysed cells served as high control. **f** TNF-α release of BMDMs 24 h after co-incubation with *C. albicans* WT (SC5314), *eed1Δ/Δ* mutant or heat-killed (HK) WT cells. BMDMs were left untreated (BMDM only) or stimulated with 100 ng/ml LPS as positive control. **g** Release of IL-6 and IL-8 by renal (A498), hepatic (HepaRG) and oral (TR146) epithelial cells 24 h after infection with *C. albicans*. Uninfected cells served as negative control. **b**, **d**, **e**, **f**, **g** Mean ±SD of four biologically independent experiments are shown. **b**, **d** In each experiment, three wells were infected with each strain and fungal survival was quantified per well. The mean of these three samples is shown as single point for the individual experiment. Data were analyzed by two-tailed student's *t*-test. *p*-values are shown in the graph. Source data are provided as a Source Data file.

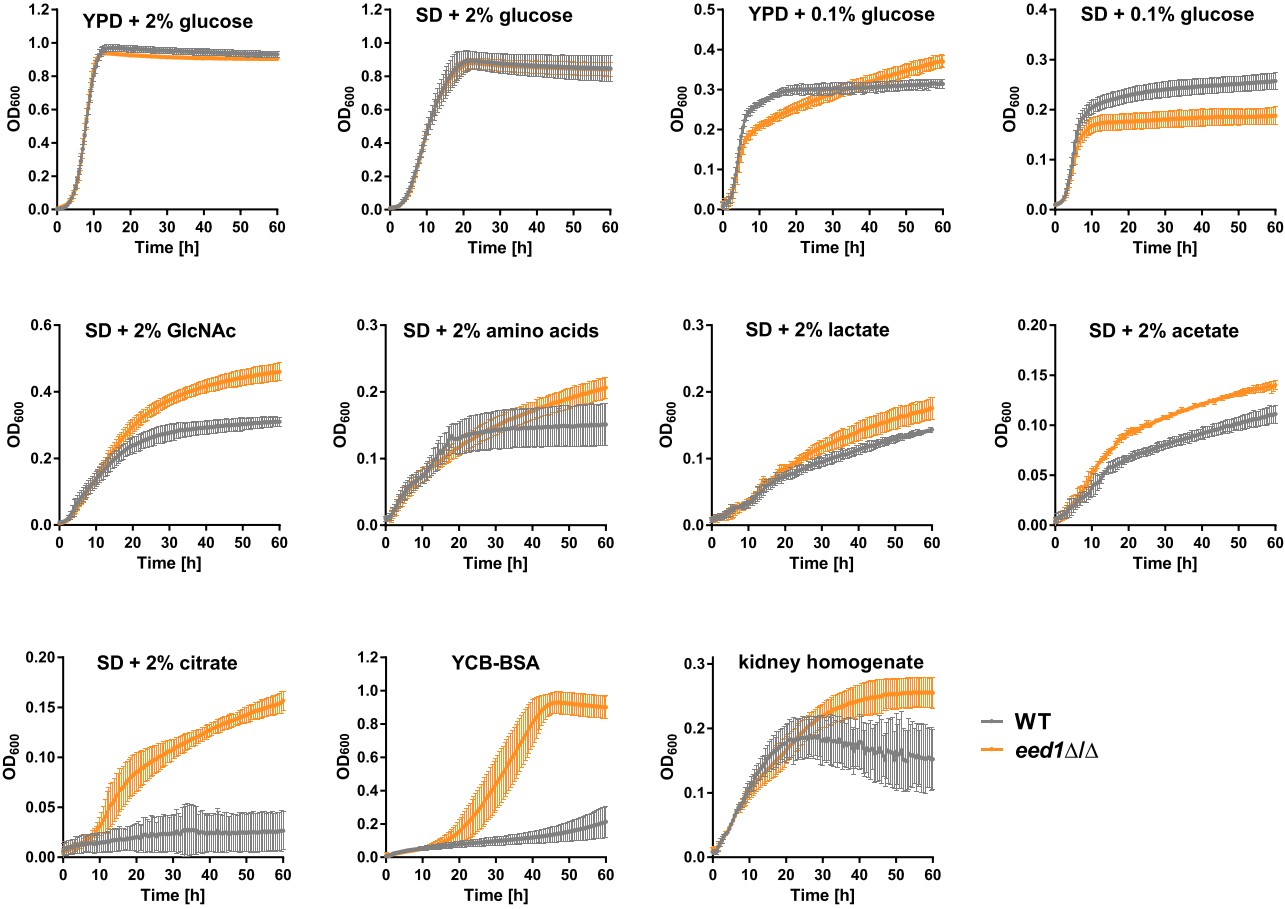

**Fig. 6 *eed1Δ/Δ* yeast have a growth advantage on physiologically relevant carbon sources and kidney homogenates and furthermore show enhanced proteolytic activity.** Growth curves of *C. albicans* in YPD with 2 or 0.1% glucose or SD medium with different concentrations of sugars or alternative carbon sources as indicated in the graphs. YCB-BSA contained 0.5% of BSA as sole nitrogen source, pH was adjusted to 4.0. For kidney homogenates, kidneys of uninfected mice were removed aseptically, homogenized and diluted to 25 mg/ml in DPBS. Growth was recorded by measuring the optical density at 600 nm in a microplate reader at 37 °C. Measurements were performed in 30 min intervals over a course of 60 h. Graphs show the mean ± SD of three to five independent biological replicates. Source data are provided as a Source Data file.

is able to adhere and invade into epithelial cells in vitro[20,50], which might be sufficient to mediate entry into renal tissue in vivo. Thus, while filamentation is required for active tissue penetration in the intraperitoneal infection model, it might be dispensable if defensive barriers are breached by direct injection into the blood stream.

Following establishment of the fungus in the kidney in immunocompetent mice, pathogenesis is thought to be driven by both direct fungus-mediated damage and immunopathology. Enhanced immunopathology could compensate for the lack of hypha-mediated fungal damage during pathogenesis; however, in the early time points following infection with an intermediate infectious dose of t-EED1+ yeast we rather observed a reduced induction of pro-inflammatory cytokines, likely as a result of the reduced capacity of the yeast to directly damage epithelial cells and macrophages and stimulate cytokine release. Macrophages have been shown to be less prone to killing by hypha-deficient strains[51] and to release less TNF-α after stimulation with *C. albicans* strains unable to germinate;[52] likewise, murine macrophages produced less TNF-α in response to the *eed1Δ/Δ* mutant. However, the amount of TNF-α in infected kidneys was not affected by absence of *EED1*-driven filamentation and the interaction with murine neutrophils in vitro was comparable for WT and *eed1Δ/Δ* mutant. Interestingly, while killing of the mutant by BMDMs was increased after 6 h of co-incubation, the relative increase in CFU was higher for the *eed1Δ/Δ* mutant than for the WT, suggesting that the increased proliferation is sufficient to compensate for the higher killing rate mediating survival and allowing for continuous organ colonization. It thus appears unlikely that the higher renal fungal burden of the yeast is solely due to reduced fungal clearance as a result of impaired recognition by and/or activation of immune cells.

We hypothesized that increased fitness mediated by better metabolic adaptation to the locally available nutrients facilitated rapid proliferation of *EED1* deficient yeast in vivo in the systemic infection model, and that the higher fungal burden in the liver after intraperitoneal infection might likewise reflect increased growth of the mutant. Upon infection *C. albicans* faces a hostile environment with varying nutritional compositions. Preferred carbon sources such as glucose are present only in low concentrations in the bloodstream (0.06-0.1%[53],) and can become scarce in microenvironments or deprived by phagocytes upon ingestion[10]. Consequently, *C. albicans* relies on the assimilation of alternative carbon sources in order to survive and proliferate within the host[54]. As Crabtree-negative yeast, *C. albicans* is able to assimilate glucose and alternative carbon sources such as amino acids, fatty acids, and carboxylic acids, at the same time[55]. This metabolic flexibility is known to increase colonization, resistance to phagocytic recognition[56] and killing, and enhances pathogenicity[57,58]. The *eed1Δ/Δ* mutant indeed showed enhanced

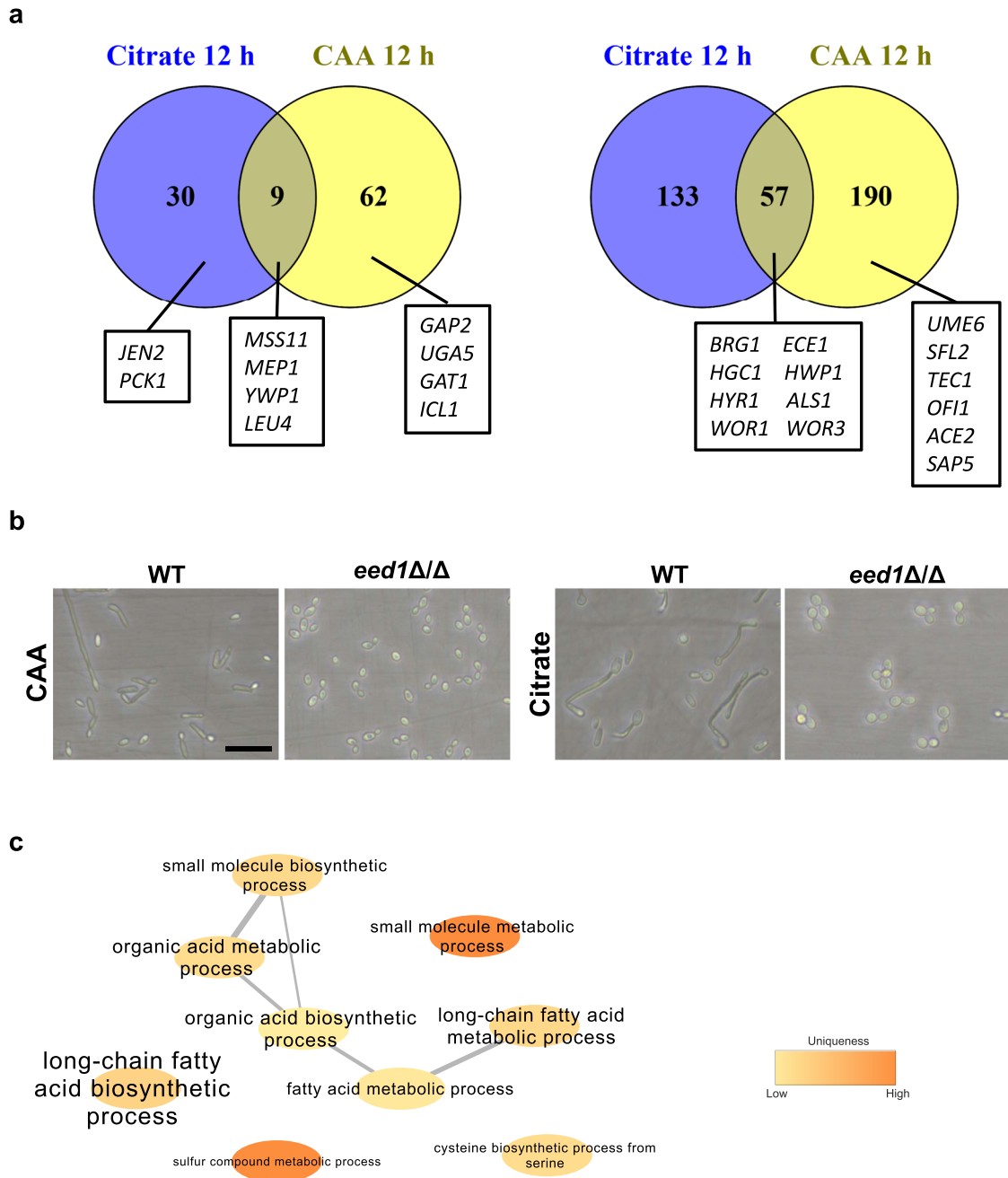

**Fig. 7 Transcriptional analysis of the *eed1Δ/Δ* mutant during growth on citrate or casamino acids (CAA) after 12 h at 37 °C. a** Venn diagrams showing the numbers of genes significantly up- or down-regulated (±log₂ 2 and adjusted *p*-value <0.05) in the *eed1Δ/Δ* mutant compared to WT (SC5314) with citrate or CAA as sole carbon source. Genes referred to in the text are highlighted. **b** Morphology of *C. albicans* WT and *eed1Δ/Δ* mutant after 12 h of growth on citrate or CAA. Representative pictures from three biologically independent experiments are shown. Scale bar represents 20 μm and applies to all images. **c** Network analysis of Gene Ontology (GO) term enrichments of significantly up-regulated genes (+ log₂ 2 and adjusted *p*-value <0.05) in *eed1Δ/Δ* mutant compared to WT after 12 h of growth on citrate. Font size represents the *p*-value (*p* < 0.1) of the GO term: big letters – small *p* value, ranging from *p* < 2.3 × 10⁻⁵ to low *p*-value *p* < 0.09 indicated by small letters. Color is representing the uniqueness, indicating how particular each term is with respect to the set of terms being evaluated (original data analyzed are available in Supplementary Data 1).

growth in kidney homogenates and in the presence of alternative carbon sources that are available in vivo, such as acetate, lactate, amino acids, citrate, and the amino sugar N-acetyl-glucosamine. Additionally, the *eed1Δ/Δ* mutant displayed an enhanced proteolytic activity, possibly supporting utilization of host proteins and immune evasion by degradation of complement proteins and antimicrobial peptides in vivo[59–61]. Whereas hypha-associated *SAPs* appeared to be down-regulated during growth on casamino acids, enhanced expression of *SAP1, SAP3, SAP7, SAP9* and

*SAP10* was observed. Since Sap1–3, Sap4–6 and Sap9–10 have different pH optima for activity, pH 3–5, pH 5–7 and pH 5–8, respectively, and in addition Saps differ in their substrate specificity[31,62], this compensatory gene expression might lead to the earlier onset of BSA utilization observed for the *eed1Δ/Δ* mutant. The increased growth on citrate is especially interesting as citrate is present in blood in a range of 0.05 to 0.3 mM, freely filtered in the glomerulus, and extensively reabsorbed in the nephrons of the kidney[63]. Therefore, citrate levels are

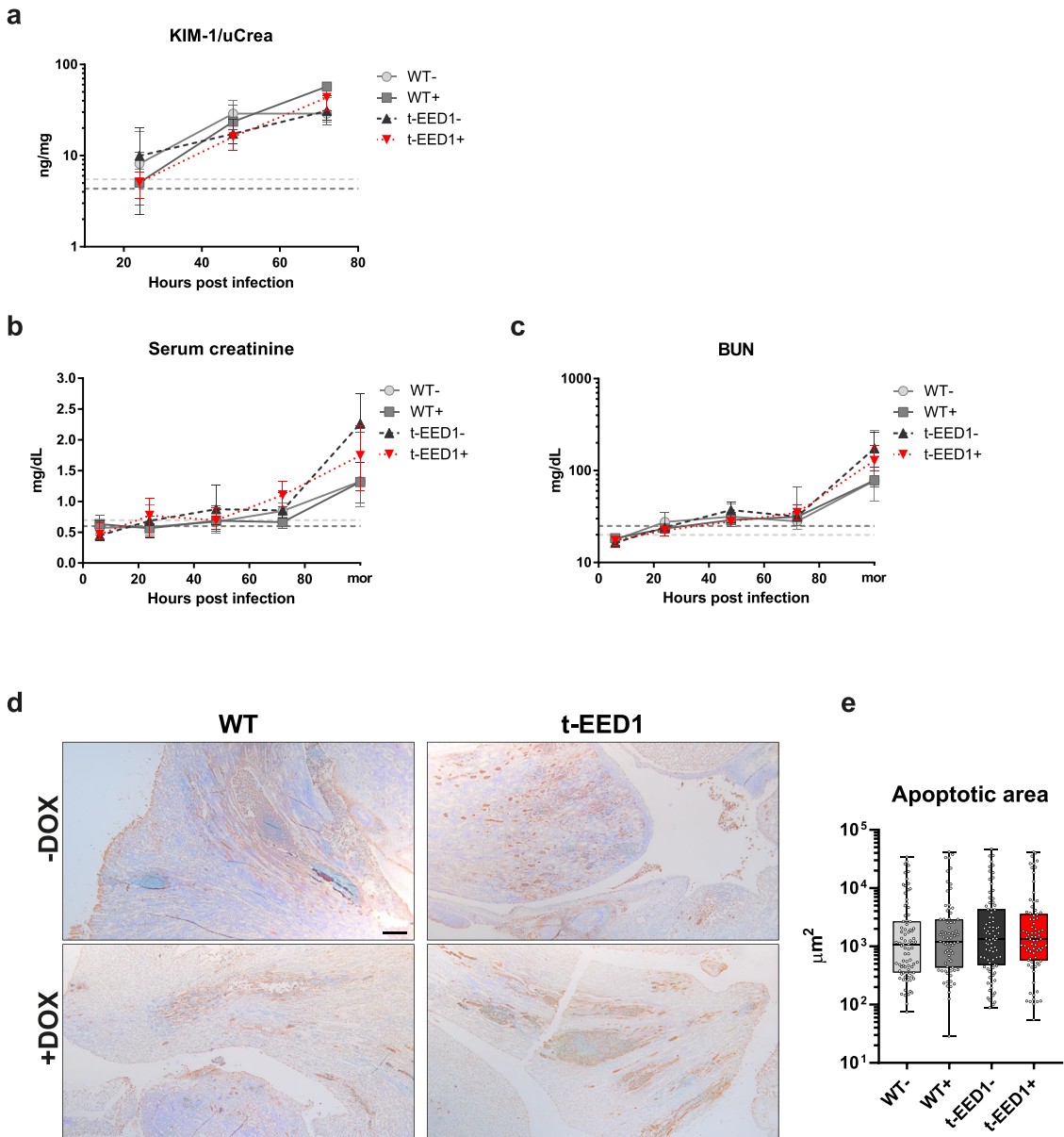

**Fig. 8 Kidneys are injured early after intravenous challenge with *C. albicans* by yeast and hyphal forms, while kidney function is not impaired within 72 h post infection. When moribund, mice developed severe renal dysfunction accompanied by renal apoptosis. a** Kidney injury was quantified by measuring and normalizing urinary KIM-1 level to urinary creatinine 24, 48 and 72 h post infection. Two independent experiments, $n = 10$ per group except for WT- 24 and 48 h and t-EED1+ 72 h: $n = 9$. Biomarkers for kidney function **b** serum creatinine and **c** blood urea nitrogen (BUN) were quantified in serum of mice 6, 24, 48, 72 h post infection and increased especially in serum of moribund (mor) mice. Two independent experiments. Serum creatinine $n = 10$, except for WT− mor, WT+ 24–72 h, t-EED1+ 24 and 48 h $n = 9$, for t-EED1− 6 h and t-EED1+ mor $n = 8$. BUN $n = 10$, except for WT− 24, 48 h and mor $n = 9$, t-EED1+ mor $n = 8$. **a–c** Dashed lines indicate median serum biomarker level of uninfected controls in the presence (+, dark gray) or absence of doxycycline (−, light gray). Data are shown as median and interquartile range. **d** Immunohistochemistry of the renal pelvis of moribund mice identified apoptotic areas stained in brown. Scale bar represents 200 μm. **e** Quantification of apoptotic areas in kidneys of moribund mice ($n = 5$ per group). Shown are apoptotic areas in μm in box-and-whiskers graph with min to max. No significant changes were observed by two-sided Mann–Whitney test. Source data are provided as a Source Data file.

approximately 3–4-fold higher in the renal cortex than in plasma. Furthermore, mice develop metabolic alkalosis early after systemic infection[48] which is known to further increase citrate concentration in the cortex, and in addition enhances renal citrate excretion 20-fold compared to the steady state[63]. Therefore, citrate could be an abundant source of carbon available for *C. albicans* in the renal cortex and tubules during the early phase of systemic infection. Thus, enhanced growth on citrate might contribute to the higher renal fungal burden in the absence of *EED1* expression. Transcriptional profiling

during growth of *C. albicans* on citrate as sole carbon source revealed only few genes that were up-regulated in the *eed1Δ/Δ* mutant compared to the WT after 12 h. These, however, included genes involved in metabolism of carboxylic acids and hence could aid in the metabolism of citrate. Similarly, upregulation of *GAT1* might be promoting growth on casamino acids. However, many metabolic enzymes are regulated by post-transcriptional modifications to respond quickly to changing environmental conditions[64,65], and it is therefore likely that the transcriptional changes described here reflect only in part the

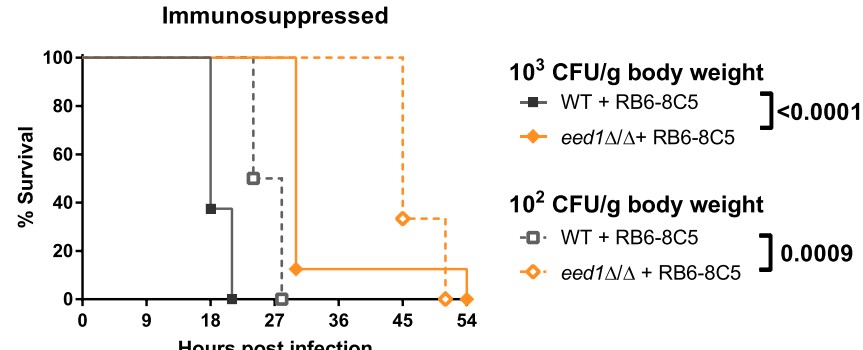

**a**

Immunosuppressed

% Survival vs Hours post infection

$10^3$ CFU/g body weight
- WT + RB6-8C5
- $eed1\Delta/\Delta$ + RB6-8C5
] <0.0001

$10^2$ CFU/g body weight
- WT + RB6-8C5
- $eed1\Delta/\Delta$ + RB6-8C5
] 0.0009

**b**

| | Uninfected | WT | $eed1\Delta/\Delta$ |
|---|---|---|---|
| Immunocompetent | | | |
| Immunosuppressed | | | |

**c**

Kidney — CFU/g (WT, $eed1\Delta/\Delta$ at 12 h, 18 h, mor)
- 12 h: p = 0.019
- 18 h: p = 0.002
- mor: p = 0.002

Liver — CFU/g
- 12 h: p = 0.01
- 18 h: p = 0.002
- mor: p = 0.002

Spleen — CFU/g
- 12 h:
- 18 h: p = 0.002
- mor: p = 0.002

**d**

KIM-1/uCrea — ng/mg
- 12 h: p = 0.01
- 18 h: p = 0.038, p = 0.026, p = 0.01
- mor: p = 0.038, p = 0.004, p = 0.01
- ctr

cellular changes leading to enhanced proliferation of the $eed1\Delta/\Delta$ mutant.

Infection with t-EED1+ yeast resulted in higher organ fungal burden, significantly more leukocytes infiltrating the kidneys and increased renal pro-inflammatory cytokine production at later time points contributing to local immunopathology[49,66]. Markers of systemic inflammation, sTREM-1 and NGAL, increased rapidly in serum of infected mice irrespective of fungal morphology. NGAL thresholds defining sepsis in humans[67] and sTREM-1 thresholds in mice[68] were reached within 24 h after infection, while damage of proximal tubular cells measured by KIM-1 became detectable slightly later (between 24 and 48 h p.i.).

Impaired kidney function determined by increased blood urea nitrogen level and serum creatinine was only evident in moribund mice, consistent with findings by Spellberg et al.[48], again without significant differences between hypha-forming and filament-deficient strains. Histologically, t-EED1+ yeast were found in large numbers mainly within tubules and in the renal pelvis; it appears possible that accumulation of yeast led to obstruction of tubules and the collecting duct system[69], increasing the intrarenal pressure and thereby, in addition to immunopathology, causes renal tissue damage. This is supported by our data showing that the area of apoptotic tissue in kidneys did not differ between mice infected with WT and t-EED1+ and suggests that the lack of

**Fig. 9 Systemic infection with the *C. albicans* eed1Δ/Δ mutant leads to delayed mortality of immunosuppressed mice despite higher fungal burden accompanied by increased kidney injury compared to mice infected with the WT.** Mice that were rendered immunosuppressed by depletion of neutrophils and monocytes using the RB6-8C5 antibody were intravenously infected with $1 \times 10^2$, $1 \times 10^3$ or $1 \times 10^4$ CFU/g body weight of *C. albicans* WT (SC5314) or *eed1Δ/Δ* mutant. **a** Survival of mice was monitored after infection with $1 \times 10^2$ CFU/g or $1 \times 10^3$ CFU/g body weight for 21 days. Survival of mice infected with $1 \times 10^2$ CFU/g ($n = 6$ per group) or $1 \times 10^3$ CFU/g body weight ($n = 8$ per group) is shown as Kaplan–Meyer curve and curves were compared using the two-sided Log-rank (Mantel–Cox) test. *p*-values are shown in the graph. **b** Representative images of PAS stained histological cross sections of kidneys from moribund immunocompetent or immunosuppressed mice infected with $10^4$ CFU/g body weight of WT or *eed1Δ/Δ* mutant. Immunocompetent and immunosuppressed uninfected control mice were sacrificed 7 d after mock infection. Black arrows point towards purple stained *C. albicans* hyphae (WT) or yeast (*eed1Δ/Δ* mutant), white arrows towards immune cells infiltrating the renal tissue. Scale bar represents 20 μm and applies to all images. **c** Organ fungal burden of immunosuppressed mice infected with $1 \times 10^2$ CFU/g body weight 12 h and 18 h post infection and when moribund (mor). **d** Quantification of kidney injury by measuring and normalizing urinary KIM-1 to urinary creatinine level. **c, d** Two independent experiments, $n = 6$, except for *eed1Δ/Δ* mutant 12 h, $n = 4$; controls, $n = 4$. Data are shown as median with interquartile range and were compared using the two-sided Mann–Whitney test. *p*-values are shown in the graph. **d** *p*-values above bars represent significant changes in comparison to the uninfected control. Source data are provided as a Source Data file.

hyphae-driven direct damage can be compensated by enhanced yeast cell growth. The exact molecular mechanisms by which absence of *EED1* influences both morphogenesis and growth on alternative carbon sources remains unknown; however, it should be noted that retention of some level of virulence in the absence of hyphal elongation is not a unique feature of *EED1* deficient strains. Homozygous deletion of *UME6*, resulting in a similar filamentation defect as observed for the *eed1Δ/Δ* mutant, leads to significantly attenuated virulence, but still causes lethal infections in mice[70]. Virulence of a *sfl2Δ/Δ* mutant was comparable to the WT despite a filamentation defect[71]. In contrast to the *eed1Δ/Δ* mutant, systemic infection with the *sfl2Δ/Δ* mutant, however, did not result in increased fungal burden, and some filamentation occurs in the absence of *SFL2*[71,72]. The role of *SFL2* for interaction with immune cells has not been investigated so far, and it thus remains unclear if differences in immunopathology contribute to the virulence of the *sfl2Δ/Δ* mutant. Furthermore, a screen of a *C. albicans* deletion library identified mutants with defects in filamentation but with unaltered infectivity;[73] a detailed analysis of these mutants might reveal additional strains for which virulence in murine systemic candidiasis models does not depend on filamentation.

Of note, significantly higher CFUs in kidneys were also observed for tet-*NRG1*- yeast on day 3 after systemic infection compared to both the parental strain (Supplementary Fig. 4d) and the same strain with doxycycline ($p = 0.001$), consistent with observations by Saville et al.[14]. However, the fungal burden of tet-*NRG1*- yeast was approximately 1-log lower than for t-*EED1*+ yeast at this time point (mean $3.19 \times 10^5 \pm 2.5 \times 10^5$ compared to $3.45 \times 10^6 \pm 1.15 \times 10^5$), and declined over time (Supplementary Fig. 4d). This indicates that while increased proliferation in vivo might be a feature shared by different filament-deficient strains, it quantitatively differs; the lower fungal burden resulting from infection with tet-*NRG1* yeast compared to t-*EED1* yeast is likely one factor contributing to the difference in virulence between these two strains. In addition to metabolic fitness, resistance to host defense mechanisms determines the extent to which *C. albicans* can proliferate in vivo. While *C. albicans* locked in the yeast form by constitutive expression of *NRG1* (tet-*NRG1*) were avirulent in immunocompetent mice in this and previous studies[14,74], the same strain was capable of inducing lethal infection in mice rendered leukopenic by combined treatment with cyclophosphamide and cortisone acetate[75]. In these mice, the yeast-locked strain reached a significantly higher fungal burden than the corresponding filamentous strain[75], which was also higher than observed in immunocompetent animals in this study and by others[14]. Furthermore, in the same study the fungal load observed in mice with different types of immunosuppression and infected with the filamentous strain was comparable at the

humane endpoint, even though this was reached at different time points after infection[75]. Similarly, MacCallum and Odds[76] observed comparable kidney burden in mice at the humane endpoint even if animals were challenged with different infectious doses and survived for a different duration. Together, this suggests that a certain number of fungal cells can be tolerated within the kidney, and that this threshold is higher for yeast than for hyphae.

Comparison of tet-*NRG1* and *EED1*-deficient strains furthermore indicates that, despite a shared morphology, different yeast strains display substantial differences in the ability to proliferate in vivo in the presence of functional innate immunity.

To our knowledge, this is the first study showing that increased metabolic fitness of *C. albicans* not only contributes to virulence in hematogenously disseminated candidiasis, but that enhanced proliferation of yeast cells can result in pathogenesis and mortality indistinguishable from infection with hyphae-forming WT and t-*EED1*− strains. Of note, the *eed1Δ/Δ* mutant caused 100% mortality in a systemic infection model in immunosuppressed mice, although delayed compared to the WT. In the absence of hypha-mediated damage and overt immunopathology, this indicates that rapid proliferation resulting in high organ fungal loads might be sufficient to drive pathogenesis. Previous studies have reported similar results: tet-*NRG1* yeast, which are avirulent in immunocompetent mice (Supplementary Fig. 4[14]), can cause lethal infection in immunosuppressed mice[75]. Whether this was due to increased fungal burden was not determined. The relevance of fungal load for pathogenesis is furthermore supported by the fact that the course of disease and rate of mortality in mice with systemic candidiasis is highly dependent on the initial infection dose[76,77]. Furthermore, by adjusting the infectious dose of different strains Odds et al. achieved a comparable mean survival time of mice infected either with *C. albicans* SC5314 or an isolate (RV4688) that displayed less filamentation in the murine kidney. Interestingly, less filamentation of RV4688 also coincided with a higher fungal burden compared to SC5314 in this study[77].

Considering that immunosuppression is a major risk factor for the development of candidemia[78] the findings in this study might provide some explanation for the virulence of non-albicans *Candida* species, such as *C. glabrata* and *C. auris*, that do not form true hyphae[79,80]. Interestingly, the yeast form is the virulent cell type in most pathogenic dimorphic fungi, such as *Histoplasma* spp. and *Blastomyces* spp., that grow as mycelia in the environment but switch to yeast upon entering the host[81]. While the pathogenicity factors and pathogenesis mechanisms employed by these yeast cells differ significantly from *C. albicans* yeast, this underscores the general concept that yeast cells are not per se less virulent than hyphae.

## Methods

**C. albicans strains, strain construction and growth conditions**. The t-EED1 strain was generated in the THE1[82] background. Therefore, the URA3-tetracycline-regulable (TR) promoter region was amplified from p99CAU1[82] using primers Eed1-TET-F and Eed1-TET-R (Supplementary Table 1). Fragments were used to replace the endogenous promotor of one allele of EED1. The second allele of EED1 was deleted using the SAT1 flipping method[83] with plasmids already containing the EED1-flanking regions used to generate the homozygous eed1Δ/Δ deletion mutant M1315[7]. Transformants were selected on YPD with 200 mg/ml nourseothricin[83] and were verified by PCR and Southern Blot analysis.

The C. albicans clinical isolate SC5314[84], the isogenic eed1Δ/Δ mutant[7], t-EED1 and the respective parental strain THE1-CIp10[18] were maintained as glycerol stocks and grown on YPD (1% yeast extract, 2% peptone, 2% glucose) agar plates. Single colonies were inoculated into liquid YPD and grown overnight at 30 °C with horizontal shaking at 180 rpm. When needed, C. albicans cells were grown to exponential phase by diluting liquid cultures to an optical density at 600 nm ($OD_{600}$) of 0.2, followed by incubation at 30 °C and 180 rpm for 3–4 h. Cultures were washed twice with phosphate buffered saline (PBS) prior to experiments.

For infection experiments, fresh C. albicans colonies were inoculated into liquid YPD and grown to late exponential phase (14–16 h) at 30 °C with horizontal shaking at 180 rpm. Doxycycline hyclate (50 μg/ml; Sigma Aldrich) was added to t-EED1 cultures to prevent hypha formation. Cells were washed twice with sterile PBS and resuspended in Dulbecco's Phosphate Buffered Saline (DPBS, Gibco), counted using a hemocytometer and adjusted to the desired concentrations in DPBS. Infectious doses were confirmed by serial dilutions and plating on YPD agar plates.

**Morphology of tetracycline-regulable strains in vitro**. To investigate fungal morphology of the tet-regulable t-EED1 strain in vitro, THE1-CIp10 and t-EED1 were streaked on YPD plates in the absence or presence of 50 μg/ml doxycycline. Plates were incubated under non-hypha-inducing conditions at 25 °C for 2 days before pictures were taken from single colonies with an inverse microscope (Axio Vert.A1; Zeiss). To test for morphology under hypha-inducing conditions, $5 \times 10^4$ cells of the strains were seeded in RPMI1640 in the absence or presence of 50 μg/ml doxycycline per well in 12-well plates and incubated at 37 °C and 5% $CO_2$. Pictures were taken after various time points by inverse microscopy.

**Growth curves**. To evaluate growth in the presence of different carbon sources SC5314 and the eed1Δ/Δ mutant were diluted to $OD_{600}$ of 0.1 in YPD medium with 2 or 0.1% glucose or in SD minimal medium (0.67% yeast nitrogen base; BD Biosciences) in the presence of 2% glucose, 0.1% glucose, 2% N-acetyl-glucosamine (GlcNAc; Sigma–Aldrich), 2% sodium-DL-lactate (Sigma Aldrich), 2% potassium acetate (Merck), 2% citric acid monohydrate (Roth) or 2% casamino acids (BD Biosciences). For testing of proteolytic activity, C. albicans strains were grown in SD medium overnight, washed twice with PBS and $OD_{600}$ was set to 0.1 in YCB-BSA (1.17% Yeast Carbon Base (BD Biosciences), 1% glucose, 0.5% BSA (Serva)), pH 4.0. For growth in kidney homogenates, kidneys of uninfected mice were removed aseptically, homogenized and diluted to 25 mg/ml in DPBS and filtered through 70 μm and 40 μm cell strainers before filter sterilization. Growth was recorded by measuring $OD_{600}$ in a microplate reader at 37 °C. Measurements were performed in 30 min intervals over a course of 60 h. Blank values were subtracted from all measurements.

**Sample preparation for RNA isolation, RNA sequencing and analysis of data**. For RNAseq, overnight cultures of C. albicans WT SC5314 and eed1Δ/Δ mutant were grown in SD with 2% glucose at 30 °C and 180 rpm (Supplementary Fig. 13a). To synchronize cultures cells were inoculated at an $OD_{600}$ of 0.1 in SD with 2% glucose and grown at 30 °C and horizontal shaking at 180 rpm. After 10 h samples for the 0 h time point were taken and cells were transferred to SD medium containing 2% of citrate (Roth) or 2% of casamino acids (CAA; BD Bacto) as sole carbon source at an $OD_{600}$ of 0.2 in individual flasks for each time point. Cells were grown at 37 °C and 180 rpm. After 2 h, 6 h and 12 h samples were removed from the cultures and cell pellets for RNA isolation were obtained by centrifugation at $20,000 \times g$ for 3 min and were immediately frozen in liquid nitrogen. Additional samples were taken at these and intermediate time points to determine the optical density at 600 nm, dry mass and morphology of fungal cells. For the determination of dry mass, nylon Whatman® membrane filters with a pore size of 0.2 μm were dried at 55 °C for 24 h in a hybridization oven, weight and placed on a bottle top vacuum filter. Cells were loaded on the membrane by filtration and washed with dd$H_2O$. After drying for additional 24 h, membranes were weighed again and dry mass was calculated. Experiments were conducted in triplicates and RNA isolation was performed as previously described[7]. In brief, pellets were resuspended in 400 μl AE-buffer (50 mM sodium acetate, 10 mM EDTA) and 40 μl 10% SDS. An equal volume of phenol/chloroforme/isoamylalcohol was added followed by incubation at 65 °C for 5 min. Homogenous solutions were frozen at −80 °C for 10 min, transferred to 65 °C for 5 min. Freezing and thawing was repeated once. Solutions were centrifuged for 10 min at $20,800 \times g$ and the upper phase was transferred into a new reaction tube. 10% volume 3 M sodium acetate (pH 5.3) and 1 volume 2-propanol were added. Precipitation of RNA was carried out for 30 min at −20 °C.

After centrifugation for 10 min at $12,000g$ the supernatant was discarded and RNA pellets were washed twice with 70% ethanol. RNA was solved in RNase free water. RNA quantity was determined with a Nanodrop ND1000 (Peqlab) and quality was assessed using an Agilent 2100 Bioanalyzer (Agilent Technologies). Library preparation and RNA sequencing was carried out at Novogene (UK) Company Limited. Sequencing was performed using an Illumina NovaSeq 6000 system to obtain 150 bp paired-end reads.

Mapping of the fastq files delivered by the company (raw data are accessible at NCBI under BioProject accession number PRJNA714826, https://www.ncbi.nlm.nih.gov/bioproject/PRJNA714826) and counting of the gene transcription reads was performed using the European Galaxy server[85] that is providing an environment and sets of tools for the following analysis steps: First, FastQC was used to assess the quality of the sequences and Cutadapt was applied. Sequences were mapped to the genome of C. albicans WT (SC5314) using the RNA-Star tool, the "length of the genomic sequence around annotated junctions" parameter was set to 149. From that analysis, bam files were constructed that were used to quantify gene counts with the FeatureCount function. The reverse stranded bam files were processed allowing for fragment counts but not multimapping. The minimum mapping quality per read was set to 10. Final count files were analyzed in R, using the Deseq2 package[86] that allows searching for differentially expressed genes (DEGs) by comparing the count tables of different conditions. For this purpose, transcription profiles of the mutant were compared to the WT at the 0 h time point, as well as for each time point in medium containing citrate or casamino acids. The principal component analysis was done using the PCA function from the Deseq2 package in R. All genes were used for the calculation. Gene Ontology (GO) term enrichment analysis was performed from significantly up- or down-regulated genes (+ or - $\log_2$ 2 and adjusted p-value <0.05) in eed1Δ/Δ mutant compared to WT using the CGD GO Term Finder[87]. Based on the analysis of GO terms with REVIGO[88] graphs were created using cytoscape[89]. Venn diagrams were created using Venny2.1 (https://bioinfogp.cnb.csic.es/tools/venny/).

**Mice**. All animal experiments were performed in accordance with European and German regulations. Protocols were approved by the Thuringian authority and ethics committee (Thüringer Landesamt für Verbraucherschutz, permit numbers: HKI-19-003, 03-007/13, 03-002/11, 03-004/15, 03-008/13). Eight- to ten-week-old female specific-pathogen-free BALB/c mice (16 to 18 g), purchased from Charles River (Germany), were housed in groups of five in individually ventilated cages at 22 ± 1 °C, 55 ± 10 % relative humidity, 12 h/12 h dark/light cycle, with free access to food and water and autoclavable mouse houses as environmental enrichment.

**Intraperitoneal and systemic infection model**. For intraperitoneal infection and survival analyses after intravenous infection with THE1-CIp10, t-EED1 and tet-NRG1, mice received drinking water containing 5% sucrose without (−) or with (+) 2 mg/ml doxycycline starting 3 days prior to infection. Water was replaced every two days. The low acceptance of doxycycline containing water resulted in a loss of body weight in the doxycycline group only, indicating possible dehydration. This could have aggravated the consequences of impaired renal function caused by systemic infection and likely explains while mice infected with THE1-CIp10 reached the humane endpoints earlier if they received doxycycline (Fig. 2c). Therefore, in further experiments mice received a diet containing 625 mg/kg doxycycline (Envigo Teklad, catalog no. TD.120769). Food was replaced every 5 days; acceptance was high, resulting in body weights comparable to the non-doxycycline groups. For the intraperitoneal infection model, mice were infected intraperitoneally with $1 \times 10^8$ CFU in 500 μl DPBS and mice were humanely sacrificed 24 h post infection. For hematogenously disseminated candidiasis mice were infected with $1 \times 10^2$ to $1.25 \times 10^5$ CFU/g body weight in 100 μl DPBS via the lateral tail vein at day 0. For survival experiments, mice were euthanized when showing signs of severe illness (details described below), and these animals are referred to as "moribund". Groups of mice (n = 5) were sacrificed 6, 24, 48 and 72 h post infection with $2.5 \times 10^4$ CFU/g body weight for analysis of fungal burden, immune cell infiltration, renal cytokines, serum and urinary marker protein progression during the acute phase of infection. Uninfected control mice received (ctr +) or did not receive doxycycline (ctr−) containing food. Time point experiments were repeated two to three times.

**Induction of neutro- and monocytopenia**. To deplete neutrophils and monocytes 100 μg of the InVivo Plus anti-mouse Ly6G/Ly6C (Gr-1) monoclonal antibody (clone RB6-8C5; BioXcell) in 100 μl DPBS were administered intraperitoneally 24 h prior to infection and every 48 h thereafter. Mice were systemically infected with C. albicans WT (SC5314) and the eed1Δ/Δ mutant using $1 \times 10^2$, $1 \times 10^3$ or $1 \times 10^4$ CFU/g body weight. Successful depletion of peripheral neutrophils (neutropenia defined as less than 200 PMNs/μl blood[90]) and monocytes was confirmed when animals reached humane endpoints by white blood cell differential count using the hematology analyzer BC-5300Vet (Mindray; Supplementary Fig. 11). Uninfected control groups (n = 2) did or did not receive RB6-8C5 and were sacrificed 7 days after they were mock-infected with 100 μl DPBS into the lateral tail vein.

**Murine model of gastrointestinal colonization and dissemination**. To avoid environmental contamination, cages, bedding, bottles and drinking water were

sterilized prior to use and mice were handled exclusively in laminar flow hoods. Mice received sucrose-containing drinking water and sucrose concentration was increased from 2.5% starting 7 d prior to infection for 2 d to 5% for 1 d. From day −4 until the end of the experiment mice received antibiotics to reduce the intestinal bacterial flora: 1500 U/ml penicillin and 2 mg/ml streptomycin were added to 7.5% sucrose-containing drinking water that was replaced daily; Mice were fed with chow containing 625 mg/kg doxycycline sterilized by irradiation. On day 0, mice were inoculated by gavage with 100 μl DPBS containing $5 \times 10^7$ C. albicans WT (SC5314) or eed1∆/∆. Mice were divided in two groups: one group was only colonized whereas in the other group dissemination was induced by injecting 200 mg/kg body weight cyclophosphamide (Endoxan, Baxter) intraperitoneally on day 7 and 10 post infection. Successful depletion of immune cells was confirmed by white blood cell differential count using the hematology analyzer. Feces were collected from individual mice on day 2, 4, 7 and 14, weighed and plated on YPD agar with or without 80 μg/ml chloramphenicol for determination of fungal and bacterial CFUs, respectively. On the end of the experiment (14 days p.i.) mice were humanely sacrificed and fungal burden were determined in liver and kidney.

**Clinical monitoring and scoring.** Body weight and body surface temperature were recorded daily. After infection the health status of the mice was checked at least twice a day. For RB6-8C5 treated immunosuppressed mice, health status was recorded every 3 h for 39 h and every 6 h thereafter. An additive clinical score was determined to evaluate disease severity. For intraperitoneal infection the following parameters were included: fur, coat and posture (normal, 0; fur mildly ruffled, 1; fur strongly ruffled, 2; fur strongly ruffled and hunched posture, 3), lethargy (absent, 0; mild, 1; moderate, 2; severe, 3), intraabdominal fibrin exudation (none, 0; single, small flocks, 1; multiple adhering flocks, removable, 2; multiple adhering flocks, removing causes damage to organ, 3), presence of other symptoms like ocular discharge, diarrhea (absent, 0; present, 1). The maximum possible score was 10. For systemic infections and the colonization and dissemination model the following parameters were included: fur (normal, 0; slightly ruffled, 1; ruffled, 2), lethargy (absent, 0; mild, 1; moderate, 2; severe, 3), body temperature (normal, 0; moderately increased, 1; increased, 2; hypothermia, 3). The maximum possible score was 8. Mice were humanely sacrificed when they reached the humane endpoints defined as (i) severe lethargy, (ii) hypothermia, or (iii) a cumulative clinical score of ≥ 5. Mice were euthanized with an overdose of ketamine (100 μl of 100 mg/ml) and xylazine (25 μl of 20 mg/ml) applied intraperitoneally followed by blood withdrawal.

**Determination of serum and urinary biomarkers.** Blood was collected by cardiac puncture (intraperitoneal infection and survival experiment) or via the vena cava inferior from mice euthanized at defined time points. Serum enzyme levels of pancreatic amylase and alanine aminotransaminase (ALT) were measured using the EuroLyser CCA 180 Vet system (QinLAB Diagnostik) according to standard methods recommended by the International Federation of Clinical Chemistry. The Mouse TREM-1 ELISA Kit (RayBiotech), DetectX® Urea Nitrogen (BUN) Detection Kit (Arbor Assays), Mouse Lipocalin-2 (NGAL) ELISA Kit (RayBiotech) and DetectX® Serum Creatinine Kit (Arbor Assays) were used to measure the respective parameters in serum of mice. Urine of mice was collected from mice euthanized 24, 48 and 72 h p.i. from moribund and uninfected mice. Either spontaneous urine was collected or gentle trans-abdominal pressure was applied onto the bladder and urine was collected using untreated glass capillary tubes. Urinary KIM-1 and creatinine levels were measured using the Mouse TIM-1 ELISA Kit (RayBiotech) and the Creatinine Parameter Assay Kit (R&D Systems), respectively. KIM-1 levels were normalized to urinary creatinine to account for differences in urinary concentration.

**Quantification of immune cells by flow cytometry.** To evaluate immune cell infiltration during the course of infection, organs were perfused with normal saline after withdrawal of blood. Organs were removed and weighed. One half of each kidney was cut into small pieces and digested in the presence of collagenase D (30 μg/ml; Sigma Aldrich) and DNase I (0.7 mg/ml; Sigma Aldrich) in RPMI (RPMI 1640; Gibco) supplemented with 10% fetal bovine serum (FBS; Bio&SELL), Pen Strep (100 U/ml Penicillin and 100 μg/ml Streptomycin; Life Technologies), and 1 mM sodium pyruvate (Gibco) for 30 min at 37 °C with moderate horizontal shaking (70 rpm). Single cells were obtained by passing the digested tissue through a 70 μm cell strainer. Cells were washed and erythrocytes were lysed by addition of red blood cell lysis buffer (0.15 mM NH₄Cl, 10 mM KHCO₃, 1 mM Na₂EDTA, pH 7.2). Remaining cells were washed, resuspended in 70% Percoll (GE Healthcare) and layered under 30% Percoll. Leukocytes were enriched by density gradient centrifugation (400g, 20 min, room temperature (RT), acceleration 1, deceleration 0). Leukocytes were collected from the interphase, washed with PBS and volumes were determined. Cells were transferred in a 96-well plate. Leukocytes were stained for flow cytometric analysis and acquired on a FACSVerse (BD Biosciences). The following antibodies were used: PerCP anti-CD45 (30-F11, BD Biosciences), APC anti-CD11b (M1/70, eBioscience), eFluor anti-CD335 (29A1.4, eBioscience), FITC anti-F4/80 (BM8, eBioscience), PE anti-CD11c (N418, eBioscience), PE-Cy7 anti-MHCII (M5/114.15.2 eBioscience), eFluor anti-Ly-6C (HK1.4, eBioscience) FITC anti-CD19 (1D3, BD Biosciences), PE-Cy7 anti-CD3e (145-2C11, eBioscience). Fc

receptors were blocked by addition of anti-mouse CD16/32 (93; BioLegend) 1:50 to the staining mixture. Dead cells were excluded from analysis using the Fixable Viability Dye eFluor® 506 (eBioscience) prior to specific antibody staining. A detailed description of the gating strategy is provided in Supplementary Fig. 19. Data were analyzed using FlowJo V.10.0.8 software.

**Histopathology and immunohistochemistry.** For histology, longitudinal sections of kidneys were fixed with buffered formalin and embedded in paraffin, cut into 3–4 μm slices, and stained with periodic acid-Schiff (PAS) staining according to standard protocols. Apoptotic cells in the kidney were detected by immunohistochemistry using the ApopTag in situ apoptosis detection kit (EMD Millipore) following the manufacturer's directions. Briefly, paraffin-embedded sections were rehydrated in Histo-Clear II (National Diagnostics) and alcohols followed by washing with phosphate-buffered saline (PBS). The sections were pre-treated with 20 μg/ml Proteinase K (Ambion) in PBS for 15 min at RT. Endogenous peroxidases were blocked by incubating slides for 15 min in 3% hydrogen peroxide. Sections were incubated with equilibration buffer (EMD Millipore) for 30 s at RT, followed by terminal deoxynucleotidyl transferase (TdT; EMD Millipore) incubation at 37 °C for 1 h. Sections were further exposed to anti-Digoxigenin (EMD Millipore, Cat Number S7100) for 30 min at RT, and the positive reaction was visualized with DAB 3, 3-diaminobenzidine (DAB) substrate (Thermo Scientific). After counterstaining the specimens with 0.5% methyl green (Sigma), they were imaged by bright field microscopy. For quantification, apoptotic areas were quantified using PROGRES GRYPHAX® software (Jenoptik).

**Determination of organ fungal burden and in vivo cytokine production.** Weighed organs were homogenized in MPO buffer (200 mM NaCl, 5 mM EDTA, 10 mM TRIS pH 8, 10% glycerol, 1 mM PMSF, 28 μg/mL Aprotinin, 1 μg/ml Leupeptin) using an UltraTurrax (Ika). Homogenates were serially diluted and plated onto YPD plates containing 80 μg/ml chloramphenicol (Roth) for enumeration of CFU. Supernatants were generated by centrifugation (1500 g, 4 °C, 15 min) and frozen at −80 °C until determination of cytokine concentrations. Cytokines were quantified using a customized ProcartaPlex™ Mix&Match Mouse 12-plex (eBioscience; cytokines that were included: GRO-alpha (KC), IFN-γ, IL-1β, IL-10, IL-12p40, IL-18, IL-6, IP-10, MCP-1, MIP-1α, RANTES, TNF-α.) The plex was performed according to manufacturer's instructions using a Luminex Magpix system (Luminex Corporation).

**Neutrophil isolation from bone marrow and differentiation of bone marrow-derived macrophages.** Bone marrow was obtained from 8–20 week old female BALB/c mice as described previously[91]. Briefly, mice were euthanized by cervical dislocation and femora, tibiae and humeri were removed and placed in RPMI supplemented with Pen Strep. Bone marrow was flushed with supplemented RPMI and single cell suspensions were obtained by continuous pipetting. Bone residues were removed by filtration through a 40 μm pore-size filter. Cells were pelleted and erythrocytes were lysed by addition of RBC lysis buffer. Cells were resuspended in Hanks' balanced saline solution without Ca and Mg (HBSS⁻; Lonza). Mature neutrophils were purified using a discontinuous Percoll gradient consisting of 52%, 69%, and 78% Percoll in HBSS⁻. Mature neutrophils were recovered from the 69%/78% interphase after centrifugation (1500 g, 4 °C, 30 min, acceleration 2, deceleration 2), washed and resuspended in HBSS⁻. Neutrophils were counted using the hematology analyzer. Purity of neutrophils was confirmed by flow cytometry to be between 89 - 95%. For differentiation into macrophages, bone marrow cells were seeded at a density of $5 \times 10^6$ cells in 175 cm² cell culture flasks in RPMI containing 10% heat-inactivated (h.i.; 30 min at 56 °C) FBS, Pen Strep and 40 ng/ml recombinant murine M-CSF (ImmunoTools). Cells were incubated in a humidified incubator at 37 °C with 5% $CO_2$ and medium was exchanged every 2–3 days. After 7 days, adherent cells were detached in RPMI + FBS by scrapping. Viable cells were counted using trypan blue exclusion and diluted to desired concentrations. For phagocytosis assays $5 \times 10^5$ neutrophils or macrophages were allowed to adhere to sterile coverslips in a 24-well plate for 1–2 h at 37 °C, 5% $CO_2$ in a humidified incubator. To increase the adherence of neutrophils, coverslips were pre-treated with 0.1% gelatin and incubated at 4 °C overnight. Wells were washed twice with PBS before seeding. For cytokine measurement, survival and damage assays $8 \times 10^4$ neutrophils or macrophages were seeded in 96-well plates in RPMI supplemented with 1% mouse serum. Cells were allowed to adhere to the substrate by culturing them for 1–2 h at 37 °C, 5% $CO_2$ in a humidified incubator prior to infection.

**Phagocytosis, survival and damage assays.** To quantify phagocytosis, cells were infected with C. albicans at a multiplicity of infection (MOI) of 1 in the presence of 1% murine serum in a total volume of 500 μl. After 1 h of co-incubation at 37 °C with 5% $CO_2$, cells were fixed with 2% paraformaldehyde. Extracellular C. albicans cells were stained with Alexa Fluor 647-conjugated Concanavalin A (Thermo Fisher Scientific) for 30 min, intra- and extracellular fungal cells were stained with Calcofluor White (Sigma-Aldrich) after permeabilization of immune cells with 0.5% Triton X-100. Coverslips were mounted with ProLong Gold antifade reagent (Thermo Fisher Scientific) and fluorescence images were recorded using the Axio Observer.Z1 (Carl Zeiss Microscopy). The phagocytic index was determined by counting the numbers of C. albicans cells phagocytosed by 100 immune cells.

Fungal survival in the presence of immune cells was determined by infecting macrophages or neutrophils with *C. albicans* (MOI1) in the presence of 1% murine serum in a total volume of 150 μl. After 2 or 6 h, immune cells were lysed by addition of 50 μl 5% Triton X-100. Fungal cells were resuspended by rigorously pipetting and lysates were diluted and plated onto YPD plates and incubated for 48 h at 37 °C. Survival rates were calculated by normalization from control wells containing no immune cells and the increase in fungal CFU was calculated by normalization to the starting inoculum for cells in the presence or absence of BMDMs. Fungal morphology was recorded by inverse microscopy using an Axio Vert.A1 microscope (Zeiss) after various time points. To quantify damage and TNF-α, BMDMs were co-incubated with *C. albicans* (MOI1) for 24 h. For total LDH release (high control), BMDMs were lysed by addition of 20 μl 5% Triton X-100, incubated for 10 min at 37 °C. Supernatants were obtained by centrifugation at 300 x *g* for 10 min. LDH was quantified using the Cytotoxicity Detection Kit (Roche) and TNF-α was quantified by ELISA (Ready-SET-Go, eBioscience) according to manufacturer's instructions.

**Cytokine and ROS production.** Macrophages and neutrophils were infected with living or heat-killed (HK; 70 °C, 10 min) *C. albicans* WT cells (MOI1) in a total volume of 200 μl. Unstimulated immune cells and cells treated with 100 nM phorbol 12-myristate 13-acetate (PMA; Sigma Aldrich) or 100 ng/ml lipopoly-saccharide (LPS; Sigma Aldrich) served as negative and positive controls, respectively. After co-incubation for 24 h at 37 °C with 5% $CO_2$ supernatants were recovered after centrifugation (1500 × *g*, 4 °C, 15 min) and TNF-α, IL-6 and IL-10 were determined by commercially available ELISA kits (Invitrogen) according to manufacturer's instructions.

Total ROS accumulation by neutrophils was quantified by luminol-enhanced chemiluminescence. Therefore, $5 \times 10^4$ freshly isolated neutrophils were seeded into white clear-bottom 96-well plates (Corning) in RPMI without phenol red (Gibco). Cells were allowed to attach for 30 min at 37 °C and 5% $CO_2$ prior to infection. Neutrophils were infected with *C. albicans* (MOI1) left untreated or were stimulated with PMA as positive control. Immediately after stimulation, 50 μl of RPMI without phenol red containing 200 mM luminol (Fluka) and 16 U horseradish peroxidase (Sigma Aldrich) were added. Luminescence was recorded every 2.5 min for 190 min at 37 °C in a Tecan Infinite microplate reader. The area under the curve was calculated with GraphPad Prism 7.

**Epithelial cell infection.** The following human epithelial cell lines were used in this study: Hepatic epithelial cells (HepaRG; Gibco) were maintained in William's Medium E with GlutaMAX and HepaRG Thaw, Plate & General Purpose Medium Supplement; renal epithelial cells (A498; DSMZ) were cultivated in Minimum Essential Medium with L-glutamine supplemented with 10% h.i. FBS; oral epi-thelial cells (TR146; Episkin) were cultivated in Dulbecco's Modified Eagle Medium (DMEM) with high glucose supplemented with 10% h.i. FBS; intestinal epithelial cells (Caco-2 clone type C2BBe1; ATCC®CRL-2102™) were maintained in DMEM supplemented with 10% h.i. FBS and 10 μg/ml human holotransferrin (Merck Millipore). Cells were cultured in a humidified incubator at 37 °C with 5% $CO_2$ under normoxic conditions (21% $O_2$) if not stated otherwise. In addition, C2BBe1 cells were cultivated under hypoxic conditions (1% $O_2$) in a temperature controlled Hypoxystation (H35, Don Whitley Scientific) to mimic physiological intestinal $O_2$ concentrations. For damage assays and the quantification of cyto-kines, cells were detached and $2 \times 10^4$ cells (TR146, A498) or $4 \times 10^4$ cells (HepaRG) were seeded in 96-well plates 2 d prior to infection. Cells were washed and infected with exponentially grown *C. albicans* strains at a MOI of 1 in a volume of 200 μl. Medium without fungal cells served as mock control. After 24 h of co-incubation, supernatants were recovered after centrifugation (200 × *g*, 5 min). To measure epithelial integrity and translocation, Corning® Transwell® polycarbonate membrane inserts with 5 μm pore size and 6.5 mm in diameter were coated with 100 μl of 10 μg/ml collagen I for 2 h at RT before they were washed twice and placed in a 24-well plate filled with 600 μl supplemented DMEM. $2 \times 10^4$ C2BBe1 cells were seeded in 200 μl supplemented DMEM in inserts and cultivated for 14 d at 37 °C, 5% $CO_2$ with 21% $O_2$ or 1% $O_2$. Medium was replaced on day 5 and every second day thereafter. Epithelial cells were infected by adding $1 \times 10^5$ *C. albicans* cells to the upper compartment and incubated for 24 h at the conditions mentioned above. To measure epithelial integrity, the trans-epithelial electrical resistance (TEER) was quantified using a chopstick electrode connected to the Epithelial Voltohmmeter EVOM2 (WPI) before and 24 h post infection. TEER measurements from inserts containing medium only served as blank values and were subtracted from all measurements. Supernatants of the upper compartment were kept for measurement of lactate dehydrogenase (LDH). To quantify the potential of the different *C. albicans* strains to translocate through the C2BBe1 cell layer, 24 h after infection the lower compartment was treated with 20 U/l zymo-lyase (Amsbio) for 2 h at 37 °C and 5% $CO_2$[22]. Detached fungal cells were plated on YPD agar and CFUs were counted. Epithelial cell damage was quantified by measurement of LDH in supernatants using the Cytotoxicity Detection Kit (Roche). Uninfected cells served as negative control. For total cells lysis (high control) 10 μl of 5% Triton X-100 were added. Human IL-6 and IL-8 were quantified by ELISA (Invitrogen) according to the manufacturer's instructions.

**Quantitative PCR.** DNA was isolated from kidneys infected with either the WT (THE1-CIp10) or t-EED1 in the presence of doxycycline. Kidneys were homogenized and centrifuged for 15 min at 1500 × *g*. DNA was extracted from pellets using the Yeast DNA Extraction Kit (Thermo Scientific) following manufacturer's instructions. For amplification of the *C. albicans* 18 S rRNA gene *RDN18* the following primers were used: sense amplification primer, 5'-GGACCCAGCCGAGCCTT-3' and antisense amplification primer, 5'-AAGTAAAAGTCCTGGTTCGCCA-3'[30]. Quantitative PCR was conducted using 1 μl of template DNA and the QPCR Mix EvaGreen (Bio&SELL) on a CFX 96 Real time System (BioRad). The following condition were used for product amplification: 95 °C for 15 min, 40 cycles of each 95 °C for 15 s, 59 °C for 15 s and 72 °C for 15 s. To confirm PCR product specificity, a melting curve was generated. The resulting Ct values were plotted against the CFU determined from the homogenized tissue.

**Statistical analysis.** GraphPad Prism 7 was used to analyze all data sets. Shown are either the mean and standard deviation (SD) or the median and the inter-quartile range as indicated in the figure legends. The two-tailed student's *t*-test or the Mann–Whitney test was used to test for statistical significances. *p*-values ≤ 0.05 were considered significant, *$p \leq 0.05$; **$p \leq 0.01$; ***$p \leq 0.001$. Survival curves were compared using the Log-rank (Mantel–Cox) test.

**Reporting summary.** Further information on research design is available in the Nature Research Reporting Summary linked to this article.

## Data availability

The RNAseq data that support the findings of this study are available at NCBI under BioProject accession number PRJNA714826 (https://www.ncbi.nlm.nih.gov/bioproject/PRJNA714826). Source data are provided with this paper.

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

## Acknowledgements

We would like to thank Birgit Weber, Nadja Jablonowski, Stephanie Wisgott and Elisabeth Rätsch for excellent technical assistance. Furthermore, we want to thank all members of the Research Group Microbial Immunology (HKI) for technical support with the animal experiments as well as the Department of Microbial Pathogenicity Mechanisms (HKI) and Franziska Gerwien for encouraging discussions. The work was financially supported by the German Research Foundation (DFG; JA1960/1-1 to IDJ; and in part through the TRR 124 FungiNet, "Pathogenic fungi and their human host: Networks of Interaction," DFG project number 210879364, Project C5 to IDJ and B2 to TD). CD was supported by the International Leibniz Research School for Microbial and Bimolecular Interactions (ILRS). M.S. was supported in part by NIH grant R00DE026856.

## Author contributions

Conception and design of the study was performed by C.D., M.P. and I.D.J. All authors contributed with data acquisition and analysis: C.D., M.P., B.S., K.S., S.R., A.E.G., T.P., S.S., R.M. and I.D.J. were involved in animal experiments; T.P. and A.G. performed epithelial co-infections; C.D. and B.S. performed flow cytometry; M.S. performed immunohistochemistry; C.D., K.S. and M.J.N. performed neutrophil experiments. C.D. and K.S. prepared samples for RNAseq. J.P.P. and T.D. performed differential gene expression analysis. Data were interpreted by C.D., M.P., B.S., T.P., M.S., S.R., A.E.G. and I.D.J. C.D. and I.D.J. wrote and all authors commented on the manuscript.

## Funding

## Competing interests

The authors declare no competing interests.
