## [Peer Review File · Nature Communications]

REVIEWER COMMENTS

Reviewer #1 (Remarks to the Author):

The authors show that *eed1* mutants of *Candida albicans* are attenuated in their ability to damage epithelial cells and cause intraperitoneal infections but were virulent in a systemic model of infection – especially at low infectious doses. This is attributed to the ability of the mutant to rapidly disseminate in the yeast form and to cause inflammatory mediated damage and to be better able to utilise a range of alternative carbon sources for growth. They conclude that their work challenges the paradigm that the hyphal phase of this fungus is essential for pathogenesis and suggests that metabolic fitness of *C. albicans* contributes to its virulence during infection.

They show that the *eed1* yeast cells accumulate at much higher levels in kidneys in the systemic model and that these yeast cells induced immune infiltrates at a much higher level but the yeast cells induced less cytokine despite this higher tissue burden. This high tissue burden is likely to be to reduced capacity for clearance and possibly enhanced ability for growth on citrate.

The arguments are well articulated and the data seem to be well presented and robust. A major positive aspect of the study is the use of multiple models to explore the *in vivo* and *in vitro* phenotype of this mutant. The primary weakness of the study is that the mechanism by which *eed1* depletion influences both yeast-hypha morphogenesis and growth on alternative carbon sources remains unknown.

It would be useful to have also used the gut model of infections, because here it is known that the ability to form hyphae is important for translocation across the gut and thus dissemination. In their intraperitoneal model, hyphae are shown to be required for tissue invasion and this would reinforce the general conclusion that hyphae are important in disease progression via tissue infiltration unless infiltration can occur by an alternative mechanism such as endothelial endocytosis. In the kidney (the primary target for invasion in the mouse systemic invasion model), it is argued that hyphae may not be absolutely required for invasion of the tissue – rather it is dependent on adherence and induced endocytosis on endothelial cells. If my interpretation of their findings is correct I think the Discussion could further clarify where hyphae are essential and less important during disease dissemination.

The authors suggest that their evidence may account for the virulence of non-filamentous *Candida* species such as *C. glabrata* and *C. auris*. They might also reflect on the fact that most pathogenic dimorphic species (*Histoplasma*, *Paracoccidioides*, *Talaromyces* etc) are only pathogenic in the yeast form.

The authors suggestion that a profusion of hyphae might trap cells locally so that so that dissemination via the yeast form is prevented. What evidence is there for this?

The authors note that having a high tissue burden can in itself potentiate disease, even if the cells are cell-for cell less virulence than the wild type. Note that it has been shown that virulence can be compensated for by adjusting the inoculum. Odds, et al. (2000). Survival in experimental *Candida albicans* infections depends on inoculum growth conditions as well as animal hosts. *Microbiology* 146, 1881- 1889). Also it has been shown in other studies that when a fungal cell type is less inflammatory that high tissue burdens can be tolerated (work by L. Romani) et al).

The authors show that the *eed1* mutant showed faster growth on YCB-BSA indicating this is more proteolytically active. How does the *eed1* mutant affect the transcript levels of SAP1-6 – are Sap4-6 that are normally associated with hypha formation misregulated in *eed1*?

The authors suggest, based on substrate utilisation experiments that the mutant is better able to adapt to a range of alternative carbon sources. A transcript profile of the mutant under repressing and non-repressing conditions would be useful to try to map these observations to possible pleiotropic changes in the transcriptome.

Minor points:

The authors focus on the work by Saville et al as supporting the hypothesis that the yeast-hypha

transition strongly contributes to *Candida* pathogenicity. Whilst this is true it would be appropriate in the Introduction make more of the other studies of monomorphic mutants that are compromised in virulence.

Line 147-152. This is a long and confusing sentence. Please break down and clarify.

Line 165. Yeasts are as efficient as hyphae to damage renal tissue *in vivo* / Yeasts are as efficient as hyphae in damaging renal tissue *in vivo*

Line 235. N-Acetyl-glucosamine / N-acetyl-glucosamine

The references need careful checking for consistency of formatting, and insertion of some missing data (e.g. some page numbers).

Reviewer #2 (Remarks to the Author):

This manuscript by Dunker and colleagues reports the novel and surprising discovery that the hypha deficient *Candida albicans eed1*Δ mutant retains virulence in the hematogenously disseminated infection model (but not in other models) and details their efforts to unravel the mechanism via which this phenomenon occurs. Overall, the experiments are for the most part thorough and the results significant enough to warrant publication in a high profile journal such as Nature Communications. However, there are several issues which I believe need addressing before the manuscript could be published.

1. One of the biggest issues revolves around the alternating use of an *eed1*Δ deletion mutant and a tetracycline regulatable *t-EED1* strain. I appreciate that using both might underscore the validity of their observations, but it disjoints the flow and complicates direct comparisons between the figures more difficult (especially given that this necessitates the use of different wild-type controls). It also seemed to confuse the authors themselves on occasion. As an example, the section describing the experiments in Fig. 4c (line 147) is headed "*t-EED1* + yeast display enhanced ..." when they were all performed with the *eed1*Δ strain.
2. In the intraperitoneal infection model, did the authors do an infection at another (lower) dose too? I think this might be relevant given that their subsequent data from the disseminated model demonstrate the importance of dosage with respect to *eed1* deleted or depleted strains. I also had trouble identifying the capsule in the accompanying histopathology pictures, an arrow indicating this would be very useful.
3. In the immunosuppressed infection data (reported in Fig 6a), did the authors perform an infection at a lower dose(s)? At the single dose used, all the animals died in a short period of time and, although the organ burden data reveals that the immunosuppressed mice infected with the *eed1*Δ mutant contain a much higher (~100-fold) fungal load (supporting a role for immunopathogenesis in these yeast form infections), these data are collected at the humane sacrifice endpoint. Perhaps infections at a lower dose, with time matched sacrifices (for organ burdens) and/or monitoring of serum creatinine and BUN (as performed in Fig. 5) would be more informative in teasing out the relative contribution of the host response to *eed1*Δ mutant virulence in immunocompetent animals.
4. The statement "Yeasts are as efficient as hyphae to damage renal tissue *in vivo*" (line 165) is not accurate because, as the authors data clearly indicate, many more of them are needed. In fact I thought the whole premise of the authors' model was that *eed1* yeast cells proliferate more rapidly *in vivo* thereby overwhelming host defenses and causing tissue damage.
5. I am a little surprised that the authors didn't provide a deeper discussion on the differences between the behavior of the *eed1*Δ mutant and that previously reported for the *sfl2*Δ and *tet-NRG1* strains. As the authors mentioned, the former strain is also virulent in the disseminated model, despite remaining in the yeast form. However, they found no difference in organ fungal burdens between the *sfl2*Δ mutant and wild-type parental strain. With regards the *tet-NRG1* strain, it too displays no difference in fungal burdens (this time between +/- doxycycline groups) and remains at a steady level in the yeast form, an observation verified here (in supplementary figure 2). Moreover, given that *tet-NRG1* yeast cells were virulent in some immunodeficient strains of mice, and not others, it's highly unlikely that simply increased fungal burden is responsible for the difference in disease outcome.

between immune-competent and deficient mice.

Other issues include:

1. What clinical score threshold did the authors use to define animals as moribund and/or needing to be euthanized?
2. In the phagocytosis and killing assays (described in Fig. S10), did the authors examine the morphology of the fungal cells? The *eed1*Δ mutant was previously reported to initiate hypha formation for several hours, prior to reverting to yeast growth. If this is also true here, this would explain the lack of difference between the wild-type and mutants. It might also suggest that assessing survival after a longer period of incubation would reveal differences, given that hypha formation has been shown to be required for macrophage escape.
3. In describing the phenotype of the *t-EED1* strain (lines 75-8), the authors describe it as hyperfilamentous in the absence of DOX. Do they mean it filaments more profusely (higher percentage of cells and/or longer filaments) following hypha induction or do they mean that it filaments inappropriately under yeast growth conditions (= constitutively filamentous)? This could be clarified by the inclusion of some microscopy of the strain grown under yeast and hypha conditions in the presence and absence of DOX (as a panel in one of the supplemental figures).
4. How do the authors reconcile the difference in survival rates between THE1-CIp10 +/-DOX groups (Fig. 2c), given that the fungal burden data (presented in Fig. 2e) shows no difference between these two groups?
5. For the fungal burden data reported in supplementary Fig. 2d, mice infected with the *t-NRG1* strain in the absence of doxycycline never become moribund so they need to define the specific time when these were presumably humanely sacrificed.
6. The statement attempting to link the lack of an immune response by epithelial cells infected with the *eed1*Δ mutant to damage (lines 141-4) is inaccurate, at least with respect to the oral TR146 cells. In these, the wild-type strain causes significantly increased damage compared to the mutant, but no increase in cytokine production.

Several minor points:

1. In describing previous virulence studies performed using morphotype locked *C. albicans* strains (line 43), the Braun et al. paper describing the characterization of the *NRG1* gene is cited, but not the companion paper published at the same time by Murad et al. In addition, the word hyphal should be replaced with filamentous, given that the *tup1*Δ mutant is constitutively pseudohyphal.
2. In several figures displaying *in vivo* data (e.g. Figs. 3, 5 and S11), I would suggest referring to the dashed lines as data from control, uninfected mice, rather than simply control (which could refer to the wild-type infections) or untreated (the addition of doxycycline is a treatment) to avoid confusion. Similarly, I presume that the control data presented in supplementary figures 5 through 9 was obtained from uninfected animals.
3. The legend for Fig. 1S states that the wild-type strain used in these experiments was BWP17. Given that the *eed1*Δ mutant was constructed in the SC5314 background, and there is no mention of BWP17 in the methods section, I assume that this is an oversight.
4. In Fig. 3S, the descriptions for panels a and b are swapped in the legend.
5. With regards the multiplex cytokine analysis, the methods section states that a custom 12-plex panel was employed but the data from only 11 are reported in the figures. What happened to the IL-10 data?
6. In several of the figures using the *tEED1* strain, the legends refer to the wild-type strain as THE1 when it should THE1-CIp10 (the parent is *ura-* and therefore avirulent in the disseminated model).
7. Use of the term human endpoint, rather than humane endpoint in several places.
8. Line 181, I believe the word should be obverse, rather than adverse in this context.

NCOMMS-20-24706-T: Point-by-point response

We would like to thank the reviewers for the critical reading and constructive comments that helped us to improve the manuscript. We have addressed all comments and provide a detailed point-to-point reply to all comments below.

Reviewer #1

The primary weakness of the study is that the mechanism by which *eed1* depletion influences both yeast-hypha morphogenesis and growth on alternative carbon sources remains unknown.

Response: We indeed do not yet know the mechanism by which Eed1 affects morphogenesis and growth on alternative carbon sources, even though we tried several approaches. As shown in Martin *et al.* 2011 (The *Candida albicans*-Specific Gene *EED1* Encodes a Key Regulator of Hyphal Extension; PLOS One) deletion of *EED1* results in down-regulation of hypha-associated genes including *UME6*; rescue of filamentation by overexpression of *UME6* suggested that both genes act in the same or converging pathways. However, according to *in silico* analysis Eed1 does not possess a DNA-binding domain, making it unlikely that it acts as a transcription factor. *In silico* analysis furthermore did not predict any other functional domain. We hypothesized that Eed1 might act by interacting with other proteins that affect transcription and tried to gain some insights by constructing different tagged versions in a heterozygous background. Only a C-terminal HA-tag was functional in our hands, but we were not able to detect the tagged protein by using antibodies directed against the tag in Western blot and fluorescence microscopy analyses. This might be due to low protein stability which was predicted by *in silico* analysis. As the classical approach to identify protein-protein interactions (or other types of molecular interactions) is pull-down experiments, we tried to express recombinant Eed1. Attempts in *E. coli* failed (also with a codon-optimized synthetic gene) as plasmids containing the *EED1* gene were not maintained in *E. coli* (several *E. coli* strains and plasmids were tested; we have unpublished evidence that the N-terminal part of the gene is toxic in *E. coli* for reasons yet unknown, and consequently cells die or plasmids are lost if the gene is transcribed even at very low levels). We therefore collaborated with groups specialized in heterologous expression in yeast systems, but these attempts likewise failed to produce recombinant *EED1*.

While therefore the exact molecular function of *EED1* remains enigmatic, we performed RNAseq analyses as suggested to gain further insights into the pathways that are affected by deletion of *EED1* and facilitate growth on alternative carbon sources. For the respective results, please see our response to this point below.

It would be useful to have also used the gut model of infections, because here it is known that the ability to form hyphae is important for translation across the gut and thus dissemination. In their intraperitoneal model, hyphae are shown to be required for tissue invasion and this would reinforce the general conclusion that hyphae are important in disease progression via tissue infiltration unless infiltration can occur by an alternative mechanism such as endothelial endocytosis. In the kidney (the primary target for invasion in the mouse systemic invasion model), it is argued that hyphae may not be absolutely required for invasion of the tissue – rather it is dependent on adherence and induced endocytosis on endothelial cells. If my interpretation of their findings is correct I think the Discussion could further clarify where hyphae are essential and less important during disease dissemination.

Response: We thank the reviewer for raising this point and have added data on the interaction with enterocytes and from a murine model of intestinal colonization and dissemination to the manuscript (new Fig. 2).

Consistent with the *in vitro* results for renal, hepatic and oral epithelial cells, deletion of *EED1* significantly reduced the ability of *C. albicans* to damage enterocytes (Fig. 2a), resulting in maintained barrier function and reduced translocation *in vitro* (Fig. 2b, c). *In vivo*, colonization of the intestinal tract was moderately increased for *eed1* Δ/Δ compared to the WT in antibiotic-treated immunocompetent mice (Fig. 2d), supporting the concept that growth in the yeast form is beneficial for intestinal colonization. Higher colonization levels were not

NCOMMS-20-24706-T: Point-by-point response

observed in mice treated with cyclophosphamide, likely because the WT colonized these mice in higher numbers compared to immunocompetent animals (Fig. 2d). The mechanisms of intestinal translocation are not clear: While Koh *et al.* (2008) found that filamentation is required in addition to mucosal barrier dysfunction and neutropenia to establish lethal translocation-induced disseminated candidiasis, the strain used in this study was the *efg1Δ/Δ cph1Δ/Δ* double mutant, which has a severe virulence defect in the i.v. model of systemic candidiasis. Thus, the lower mortality observed with that strain might be a consequence of the general virulence defect, rather than reduced translocation. Furthermore, basal dissemination from the gut in immunocompetent animals has been observed by others, and was found to be independent of filamentation (Vautier *et al.* 2015); indirect transport by dendritic cells or through M cells has been proposed as the underlying mechanism. We observed basal dissemination for the WT but not for the *eed1Δ/Δ* mutant (Fig. 2f). It appears possible that the *eed1Δ/Δ* mutant interacts differently with M cells or DCs, which might explain lower translocation, but we deemed it beyond the scope of this study to further investigate this hypothesis. Dissemination of the *eed1Δ/Δ* mutant to internal organs was however observed upon cyclophosphamide treatment (Fig. 2f), which resulted in a significantly higher fungal burden in kidneys compared to the WT. While one could interpret these findings in such a way that impaired barrier function by mucosal damage allows yeast to disseminate without the need for filamentation, the higher fungal load could also be the result of increased proliferation in this organ. Thus, we do not think that the data allows any conclusion on the role of filamentation for dissemination from the gut. Of note, we did not observe any symptoms of disseminated disease in the colonized cyclophosphamide-treated mice, suggesting that our cyclophosphamide treatment was not sufficient to induce the level of immunosuppression and/or intestinal damage required for translocation of a high enough number of fungal cells to cause disease. Consistent with this interpretation is the finding that the number of fungal cells (WT and mutant) in kidneys of these animals was several log levels lower than in animals that were systemically infected and showed clinical symptoms.

In addition to presenting the additional data in the results section, we have added a paragraph in the discussion, emphasizing the role of hyphae for tissue invasion and discussing the findings in the colonization model (lines 303-324). We also added a sentence at the end of the first paragraph of the discussion of the systemic model to clarify the putative role of the different morphologies in the different infection models: “Thus, while filamentation is required for active tissue penetration in the intraperitoneal infection model, it might be dispensable if defensive barriers are breached by direct injection into the blood stream.”

The authors suggest that their evidence may account for the virulence of non-filamentous *Candida* species such as *C. glabrata* and *C. auris*. They might also reflect on the fact that most pathogenic dimorphic species (*Histoplasma*, *Paracoccidioides*, *Talaromyces* etc) are only pathogenic in the yeast form.

Response: The reviewer raises here another interesting point. While the pathogenicity mechanisms of *Candida* and dimorphic fungi differ in many ways, the fact that the yeast but not the hyphal form of dimorphic fungi is virulent certainly underscores that yeast cells are not per se less virulent than hyphae. We have added two sentences towards the end of the discussion to mention this (lines 463-468 in the manuscript with track changes).

The authors suggestion that a profusion of hyphae might trap cells locally so that so that dissemination via the yeast form is prevented. What evidence is there for this?

Response: We are not sure to which part of the manuscript this comment refers to – we did not state that hyphae might trap cells and thereby prevent dissemination via the yeast form. In the description of the results obtained with the intraperitoneal model we however made the following statement: “Histological analysis showed t-EED1+ yeast that were trapped below the liver capsule whereas WT and t-EED1- hyphae invaded deeply into liver parenchyma (Fig. 1c).” lines 80/81 of the original manuscript, and a similar statement in the discussion (lines 194 of the original manuscript). These sentences aimed to describe the fact that only hypha-producing strains invaded deeply into the liver tissue, while strains in the yeast morphology were only able to invade the organ superficially. To avoid misunderstandings, we have removed the term “trapped” from both sentences (lines 88 and 301 in the manuscript with track changes).

NCOMMS-20-24706-T: Point-by-point response

The authors note that having a high tissue burden can in itself potentiate disease, even if the cells are cell-for cell less virulence than the wild type. Note that it has been shown that virulence can be compensated for by adjusting the inoculum. Odds, et al. (2000). Survival in experimental *Candida albicans* infections depends on inoculum growth conditions as well as animal hosts. *Microbiology* 146, 1881- 1889). Also it has been shown in other studies that when a fungal cell type is less inflammatory that high tissue burdens can be tolerated (work by L. Romani) et al).

Response: We thank the reviewer for drawing our attention to these important papers. Indeed, the fact that the course of disease and rate of mortality in mice with systemic candidiasis is highly dependent on the initial infection dose supports a role of fungal load in pathogenesis. Maybe even more importantly, the work by Odds et al. also showed that by adjusting the infectious dose comparable mean survival times of mice infected either with *C. albicans* SC5314 or an isolate that displayed less filamentation in the murine kidney can be achieved. In addition, a higher fungal burden was observed for the less-filamentous strain, which supports the idea that also yeast cells can drive pathogenesis if present in sufficient numbers. We therefore mention these findings in the revised discussion of our manuscript (lines 455-460 in the manuscript with track changes).

Regarding studies showing that fungal cell type that are less inflammatory are tolerated at high tissue burdens, we were unfortunately unable to find appropriate literature despite going through the work by Luigina Romani et al.. If the reviewer could direct us to appropriate papers, we'd be happy to include this point.

The authors show that the *eed1* mutant showed faster growth on YCB-BSA indicating this is more proteolytically active. How does the *eec1* mutant affect the transcript levels of *SAP1-6* – are *Sap4-6* that are normally associated with hypha formation misregulated in *eec1*?

Response: In an earlier study we compared gene expression of the *eed1* Δ/Δ mutant and WT under hypha-inducing conditions *in vitro* (growth on plastic in RPMI medium at 37 °C and 5% CO₂) and found no significant difference for expression of *SAP1-6* (Martin *et al.* 2011, Supplement). In the same study, we analyzed the expression of genes during infection of a reconstituted human oral epithelium and *SAP1-6* were not differentially expressed compared to the common control culture (SC5314 grown in YPD at 37°C to mid-log phase). In the new RNAseq data we found evidence that during growth on citrate, and to a greater extent during growth on casamino acids (CAA) the hyphal-associated genes *SAP4-6* (Naglik *et al.* 2004) were mainly down-regulated in the *eed1* Δ/Δ mutant compared to WT (SC5314), consistent with the observed morphology. In contrast, enhanced transcription of the yeast-associated *SAP1* and *SAP3* (Naglik *et al.* 2004) was observed as well as an increased expression of *SAP7* in CAA medium between 2 and 12 h of growth. Additionally, transcript levels of *SAP9* and *SAP10* were moderately increased at later time points. In citrate *SAP3*, *SAP7* and *SAP9* were up-regulated occasionally after 2 and 6 h of growth. Therefore, the regulation of *SAPs* in the *eed1* Δ/Δ mutant differs substantially from the profile expressed by the WT. In consequence, it is likely that during growth in YCB-BSA expression of *SAPs* varies among these strains. Since *Sap1-3*, *Sap4-6* and *Sap9-10* have different pH optima for activity, pH 3-5, pH 5-7 and pH 5-8, respectively and in addition *Saps* differ in their substrate specificity (Naglik *et al.* 2004, Schild *et al.* 2011), this might explain the differences in BSA utilization observed in Fig. 6. This information is presented in Supplementary Fig. 16b and discussed in lines 372-377 of the revised manuscript with track changes.

The authors suggest, based on substrate utilisation experiments that the mutant is better able to adapt to a range of alternative carbon sources. A transcript profile of the mutant under repressing and non-repressing conditions would be useful to try to map these observations to possible pleiotropic changes in the transcriptome.

Response: We fully agree with the reviewer and therefore performed RNAseq of the WT and *eed1* Δ/Δ mutant during growth on citrate and casamino acids (CAA). The new results show relatively few differentially expressed genes (DEGs) between WT and mutant in the different conditions. DEGs up-regulated in the mutant do however include genes involved in utilization of organic acids (growth on citrate) and permeases/transporters (growth on amino acids) that contribute to utilization of amino acids as carbon

NCOMMS-20-24706-T: Point-by-point response

sources. Genes associated with filamentation and expression of hypha-associated genes (HAGs) were generally down-regulated at those time points in which the mutant grew as yeast while filaments were present in WT cultures, indicating that the morphology-dependent transcriptional profile was not grossly dysregulated. Importantly, besides *GAT1* (a global regulator of nitrogen utilization) we did not identify any transcription factor directly involved with metabolism to be differentially expressed. Since metabolism is not only regulated on the transcriptional level but also (and likely to a higher extent) by posttranslational modifications, extensive proteome analyses and analyses of metabolic fluxes would likely be necessary to obtain a more comprehensive picture of the differences between the mutant and the WT – this would however require significant additional work which we consider to perform in the future but deem to be outside the aim of this study.

The new results are presented in Supplementary Fig. 16a,c and discussed in lines 385-392 of the revised manuscript with track changes.

The authors focus on the work by Saville et al as supporting the hypothesis that the yeast-hypha transition strongly contributes to Candida pathogenicity. Whilst this is true it would be appropriate in the Introduction make more of the other studies of monomorphic mutants that are compromised in virulence.

Response: We fully agree and have added additional studies to the relevant part of the introduction: “Similarly, other *C. albicans* mutants locked in the yeast form, such as the *cph1Δ/Δ efg1Δ/Δ* double and the *hgc1Δ/Δ* mutant have been shown to be avirulent or strongly attenuated in virulence^{10,11}. On the other hand, mutants locked in the filamentous form like *tup1Δ/Δ* or *nrg1Δ/Δ* are less virulent as well^{12, 13, 19}, implying that morphological plasticity is essential for virulence in murine disseminated candidiasis.” (lines 55-58 in the manuscript with track changes)

Line 147-152. This is along and confusing sentence. Please break down and clarify.

Response: We have broken down and rephrased the sentence to increase clarity (lines 190-195 in the manuscript with track changes).

Line 165. Yeasts are as efficient as hyphae to damage renal tissue in vivo / Yeasts are as efficient as hyphae in damaging renal tissue in vivo

Response: The heading was changed to “High numbers of yeast cells result in renal damage *in vivo*.” (line 259 in the manuscript with track changes) in response to comment 4 made by Reviewer #2.

Line 235. N-Acetyl-glucosamine / N-acetyl-glucosamine

Response: We changed N-Acetyl-glucosamine to N-acetyl-glucosamine throughout the manuscript.

The references need careful checking for consistency of formatting, and insertion of some missing data (e.g. some page numbers).

Response: We apologize for our carelessness regarding the formatting of the references and thank the reviewer for pointing this out. All references were screened for errors and corrected if necessary.

Reviewer #2

NCOMMS-20-24706-T: Point-by-point response

1. One of the biggest issues revolves around the alternating use of an *eed1*Δ deletion mutant and a tetracycline regulatable t-EED1 strain. I appreciate that using both might underscore the validity of their observations, but it disjoins the flow and complicates direct comparisons between the figures more difficult (especially given that this necessitates the use of different wild-type controls). It also seemed to confuse the authors themselves on occasion. As an example, the section describing the experiments in Fig. 4c (line 147) is headed “t-EED1+ yeast display enhanced ...” when they were all performed with the *eed1*Δ strain.

Response: We agree with the reviewer that using different strains complicates the reading flow. We think it is necessary, however, to show data with mutants generated by both approaches for the following reasons:

We initially tested the *EED1* deletion mutant generated in the BWP17 background and the corresponding complemented strain (Zakikhani *et al.*, 2007) in the systemic mouse model. To our surprise, the complemented strain, but not the mutant, was significantly attenuated (data not included in the manuscript). Accidental swapping of the two strains was excluded by analyzing colonies recovered from infected mouse tissue by both PCR and filamentation assays. Construction of a new complemented strain in the BWP17 background was hampered by the fact that cloning plasmids containing the *EED1* gene were not maintained in *E. coli* (several *E. coli* strains and plasmids were tested; we have unpublished evidence that the N-terminal part of the gene is toxic in *E. coli* for reasons yet unknown, and consequently cells die or plasmids are lost if the gene is transcribed even at very low levels). Attempts to generate a complementation cassette by cell-free PCR-based approaches were impaired by frequent occurrence of point mutations, even if high-fidelity polymerases were used. Therefore we decided to generate the t-EED1 strain in which gene expression could be manipulated in either direction, thereby circumventing the necessity of a complemented strain to control for pleiotropic effects caused by the mutagenesis and potentially affecting virulence. We consider it to be an additional advantage that this strain can be compared directly with the well described tet-*NRG1* strain (Saville *et al.*, 2003).

A disadvantage of the t-EED1 strain is that it requires the use of doxycycline (or derivatives) to down-regulate *EED1* – this increases the number of experimental groups. After confirming that deletion of *EED1* in the SC5314 background yielded results comparable to the t-EED1 strain in the presence of doxycycline *in vivo* and in *in vitro* filamentation assays, we thus decided to use this strain for further experiments, even though a complemented strain is not available for the same reasons as explained above.

Given that it was surprising that a strain with a strong filamentation defect remains virulent in the systemic candidiasis model, we think that it is important to demonstrate that this phenotype is consistent in mutant generated by different approaches. To avoid confusion for the readers as much as possible, we decided to use different colors for the two types of mutants in the revised manuscript (red for the t-EED1 strain with doxycycline, orange for the deletion mutant). Together with the two different designations this will hopefully improve clarity.

Regarding Fig. 4c (line 147) we apologize for the oversight and have corrected this mistake.

2. In the intraperitoneal infection model, did the authors do an infection at another (lower) dose too? I think this might be relevant given that their subsequent data from the disseminated model demonstrate the importance of dosage with respect to the *eed1* deleted or depleted strains.

Response: We thank the reviewer for this suggestion. However, the attenuated virulence phenotype of *EED1* depleted (Fig. 1) or *EED1* deleted cells (Fig. S1) at the given infectious dose in this model led to no significant difference between mice infected with the mutant and uninfected control mice for most parameters (serum markers for liver and pancreatic damage), and only a low clinical score. Thus, infection with a lower dose would not provide further insights on the attenuation of the mutant and might lead to organ damage levels in the *C. albicans* WT infected group that are too low for robust statistical analysis when compared to the uninfected controls. It might be possible to circumvent such issues by analyzing the mice at later time points – this is however not covered by our animal license and thus would require a set of experiments to establish appropriate doses and time points while avoiding a level of suffering that is deemed unacceptable by our ethics committee.

NCOMMS-20-24706-T: Point-by-point response

Another issue that would be interesting to address in the i.p. model, and for which the use of different infection doses might yield relevant insights, is the question whether the increased proliferation of the mutant observed in the systemic model also occurs in the peritoneal cavity. We have quantified CFU in the i.p. infection experiments already performed, and observed a significant (approx. 10-fold) difference between WT and mutant for the fungal burden of the liver. This data has been added to the revised manuscript (Fig. 1e and lines 93-97) and is discussed in the context of the i.p. model (lines 358-360), as it implies that the attenuation observed in this model (especially regarding the lower level of serum enzymes indicating liver damage) is not due to reduced fungal burden. This data also indicates that the increased proliferation of the *EED1* mutant occurs not only after systemic infection but also in the intraperitoneal model (discussed in lines 338-340).

We also performed peritoneal lavage before harvesting abdominal organs for further analysis, and plated the lavage fluid to assess if the fungal load in the peritoneal cavity differs. The CFU in the lavage, however, displayed very high variation between animals of the same infection group (up to 100-fold). This is likely a technical issue, as sampling of the complete peritoneal cavity, including the surface of all intestinal organs is tricky, and we have not yet found a reliable way to solve this. Therefore, this data is not included.

Thus, we have added existing CFU data (liver) to the revised manuscript, but decided not to perform additional intraperitoneal infections and hope that the reviewer understands our decision.

I also had trouble identifying the capsule in the accompanying histopathology pictures, an arrow indicating this would be very useful.

Response: Since the capsule is very thin and partly disrupted by *C. albicans* it is indeed hard to see. We thus added arrows pointing out the liver surface (capsule) in Fig. 1c.

3. In the immunosuppressed infection data (reported in Fig 6a), did the authors perform an infection at a lower dose(s)? At the single dose used, all the animals died in a short period of time and, although the organ burden data reveals that the immunosuppressed mice infected with the *eed1Δ* mutant contain a much higher (~100-fold) fungal load (supporting a role for immunopathogenesis in these yeast form infections), these data are collected at the humane sacrifice endpoint. Perhaps infections at a lower dose, with time matched sacrifices (for organ burdens) and/or monitoring of serum creatinine and BUN (as performed in Fig. 5) would be more informative in teasing out the relative contribution of the host response to *eed1Δ* mutant virulence in immunocompetent animals.

Response: We thank the reviewer for these suggestions and agree that analysis of samples from identical time points is necessary for comparison of the fungal burden. Thus, we performed additional infection experiments using immunocompromised mice and a 10-fold lower infectious dose (1×10^2 CFU/g body weight). The new data confirmed that it takes significantly longer for neutropenic mice infected with the *eed1Δ/Δ* mutant to reach the humane endpoints (time to humane endpoints: WT 18 h to 26 h; *eed1Δ/Δ* mutant 30 h to 45 h; Fig. 9a). Furthermore, analysis of immunosuppressed mice infected with this lower dose and sacrificed 12 h and 18 h post infection, respectively, revealed that the fungal burden of the *eed1Δ/Δ* mutant was significantly higher than that of the WT at matched time points in liver and kidney (Fig. 9c). Kidney damage (assessed by urinary KIM-1/creatinine ratio) and kidney function (BUN) were also analyzed in these animals. BUN did not significantly increase, likely due to the limited duration of the experiment that did not allow for significant accumulation of urea even if kidney function was impaired in moribund mice (Supplementary Fig. 18b). A significant increase in urinary KIM-1 was observed in mice infected with either WT or *eed1Δ/Δ* (Fig. 9d). For the *eed1Δ/Δ* mutant, urinary KIM-1 steadily increased, supporting that high yeast burden is sufficient to induce renal tissue damage. However, even though urinary KIM-1 was significantly higher in mice challenged with *eed1Δ/Δ* compared to the WT 18 h p.i., these mice reached the humane endpoints later (Fig. 9a), suggesting that not only yeast-driven renal damage but also the host response contributed to the full virulence of the mutant observed in immunocompetent mice. These results are presented in lines 279-290 in the results part of the revised manuscript.

NCOMMS-20-24706-T: Point-by-point response

4. The statement “Yeasts are as efficient as hyphae to damage renal tissue *in vivo*” (line 165) is not accurate because, as the authors data clearly indicate, many more of them are needed. In fact I thought the whole premise of the authors’ model was that *eed1* yeast cells proliferate more rapidly *in vivo* thereby overwhelming host defenses and causing tissue damage.

Response: We agree that the heading was misleading and have rephrased it to “High numbers of yeast cells result in renal damage *in vivo*.” (line 259 in the manuscript with track changes)

5. I am a little surprised that the authors didn’t provide a deeper discussion on the differences between the behavior of the *eed1Δ* mutant and that previously reported for the *sfl2Δ* and *tet-NRG1* strains. As the authors mentioned, the former strain is also virulent in the disseminated model, despite remaining in the yeast form. However, they found no difference in organ fungal burdens between the *sfl2Δ* mutant and wild-type parental strain. With regards the *tet-NRG1* strain, it too displays no difference in fungal burdens (this time between +/- doxycycline groups) and remains at a steady level in the yeast form, an observation verified here (in supplementary figure 2). Moreover, given that *tet-NRG1* yeast cells were virulent in some immunodeficient strains of mice, and not others, it’s highly unlikely that simply increased fungal burden is responsible for the difference in disease outcome between immune-competent and deficient mice.

Response: We initially did not include a deeper discussion on *NRG1* and *SFL2* mutants for the sake of brevity, but expanded the discussion in this regard in the revised manuscript.

Regarding the *SFL2* deletion strain (Spiering *et al.* 2010), it should be noted that contrary to the statement in the abstract of Spiering *et al.* 2010 (“ $\Delta\Delta sfl2$ strain-infected kidney tissues contained only yeast cells”), the histology shown in Fig. 8 of the paper shows some filamentous structures in kidneys of mice infected with the null mutant (see image below from Spiering *et al.* 2010). This is also acknowledged by the authors in the paper: “Interestingly, histology of kidney sections obtained from the 3- and 28-day infection models revealed that the $\Delta\Delta sfl2$ strain was defective in hypha formation, growing almost exclusively as yeast cells with only a few short filaments (Fig. 8).”

[Redacted]

Fig. 8, Spiering *et al.* 2010, Eukaryotic Cell. Panels E and F display kidney sections of mice infected with the *SFL2* null mutant. (<https://ec.asm.org/content/9/2/251>)

This is consistent with our own findings (not mentioned in the manuscript), that this mutant (which we tested in collaboration with the Hube lab who received it from the Sullivan lab and was confirmed by PCR by us to be a null mutant) does produce filaments in some *in vitro* conditions (though to a lesser extent than the wildtype):

NCOMMS-20-24706-T: Point-by-point response

RPMI1640, 37 °C, 5% CO₂, 12-well plate

Furthermore, work by the Edgerton lab (McCall *et al.*, 2018, Plos Path) also shows filamentation of a *SFL2* deletion mutant (Fig. 5) in liquid media under static and flow conditions and on epithelial cells. Of note, this study also assayed growth in RPMI, as we show above, and observed filamentation.

Thus, while deletion of *SFL2* without doubt leads to reduced filamentation, the mutant retains some ability to filament and to form visible filamentous structures *in vivo*. This might, at least in part, explain retained virulence without increased fungal burden. Other aspects that we addressed in our work for *EED1* (interaction with immune cells) have to the best of our knowledge not yet been investigated for *SFL2*, and might additionally contribute to virulence of this strain.

This is summarized in the revised discussion as follows: “Virulence of a *sfl2Δ/Δ* mutant was comparable to the WT despite a filamentation defect. In contrast to the *eed1Δ/Δ* mutant, systemic infection with the *sfl2Δ/Δ* mutant, however, did not result in increased fungal burden, and some filamentation occurs in the absence of *SFL2*. The role of *SFL2* for interaction with immune cells has not been investigated so far, and it thus remains unclear if differences in immunopathology contribute to the virulence of the *sfl2Δ/Δ* mutant.” (lines 413-417 in the manuscript with track changes).

Regarding the tet-*NRG1* strain, two publications by Saville *et al.* (2003, Eukaryotic Cell, and 2008, Infection and Immunity) do contain valuable information that we discuss in the revised manuscript in the context of our findings:

1) The paper from 2003 analyzed fungal burden 6 h after infection with 5×10^6 cells (approx. 2.5×10^5 /g body weight, assuming an average mouse weight of 20 g) and 3 days after infection with 5×10^5 cells (approx. 2.5×10^4 /g body weight, and thus comparable with our data presented in Supplementary Figure 2 of the original manuscript, new Supplementary Figure 4). While no difference in fungal burden was observed 6 h after infection, the fungal load of the tet-*NRG1* strain was significantly higher in kidney and spleen 3 days after

NCOMMS-20-24706-T: Point-by-point response

infection (Saville *et al.* 2003, Fig. 5). This is consistent with our findings for 3 days after infection presented in the supplement (note that the statistical analysis presented in our manuscript is based on comparison between mutant and parental strain, but we have re-analyzed the data for tet-*NRG1* +/- doxycycline for comparison to the data by Saville *et al.*). This might indicate better proliferation of the tet-*NRG1* yeast cells. However, fungal burden in internal organs of mice infected with tet-*NRG1* in the absence of doxycycline declined over time in our study, indicating that the increase in fungal burden is only transient. Even more importantly, the fungal burden in kidneys 72 h after infection was 1-log lower for the tet-*NRG1* strain in the absence of doxycycline (mean 3.2×10^5 /g) compared to t-*EED1* yeast (mean 3.5×10^6 /g). This indicates that while better proliferation *in vivo* might be a feature shared by different strains locked in the yeast form, it is not a universal feature and significant quantitative differences do exist, and this in turn might explain, at least in part, the virulence difference between tet-*NRG1* and t-*EED1* yeast cells. We have added the following sentences to the revised discussion to address this point: "Of note, significantly higher CFUs in kidneys were also observed for tet-*NRG1* yeast (in the absence of doxycycline) on 3 day after systemic infection compared to both the parental strain (Supplementary Fig. 4d) and the same strain with doxycycline ($p=0.001$), consistent with observations by Saville *et al.* {Saville, 2003 #18}. However, the fungal burden of tet-*NRG1* yeast was approximately 1-log lower than t-*EED1* yeast at this time point (mean $3.19 \times 10^5 \pm 2.5 \times 10^5$ compared to $3.45 \times 10^6 \pm 1.15 \times 10^5$), and declined over time (Supplementary Fig. 4d). This indicates that while increased proliferation *in vivo* might be a feature shared by different filament-deficient strains, it quantitatively differs; the lower fungal burden resulting from infection with tet-*NRG1* yeast compared to t-*EED1* yeast is likely one factor contributing to the difference in virulence between these two strains." (lines 421-429 in the manuscript with track changes).

2) An alternative explanation for the transiently higher fungal burden for tet-*NRG1* yeast is reduced killing by immune cells. While Saville *et al.* did not quantify immune cells in infected tissue, histological analysis in the paper from 2008 found differences in immunocompetent BALB/c mice infected in the presence or absence of doxycycline (Fig. 3): Numerous neutrophils and few macrophages were found to surround hyphae in the presence of doxycycline (Fig. 3A), while in the absence of doxycycline infiltrates around yeast cells were composed of mainly by macrophages and lymphocytes at the end of the experiment (Fig. 3B). This again differs from our observation with the *EED1* mutant, for which neutrophil recruitment was similar to the WT. This again supports the notion that yeast morphology *per se* is not predicative for fungal behavior in the host and interaction with immune cells. This is actually not surprising given that most yeast-locked mutants were generated by deletion of transcription factors that not only affect morphogenesis but other cellular functions as well.

3) The reviewer correctly points out that tet-*NRG1* yeast cells were virulent in some immunodeficient mice – this was specifically the case for animals treated with both cyclophosphamide and cortisone acetate, while the strain was unable to induce lethal infection in mice deficient for B cells, T cells, or with C5 deficiency (Saville *et al.*, 2008). In their publication, the authors do not present analyses of the immune cells present after cyclophosphamide-cortisone acetate treatment, but since this treatment has been frequently used in studies of invasive aspergillosis, it can be inferred from these studies that the treatment renders mice leukopenic, and thus severely immunocompromised. In addition, a comparatively high infectious dose (1.7×10^6 cells/mouse, approx. 8.5×10^4 cells/g body weight) was used on purpose to achieve lethality: "It should be noted that the experimental design for all of the above experiments was selected to maximize any chances of observing lethality associated with a yeast cell-only infection (in the absence of DOX)." Mice succumbed to the infection within 2 days. In our model of antibody-mediated depletion of neutrophils and inflammatory macrophages, which reflects some but not all aspects of leukopenia, we observed 100% mortality with substantially lower infectious doses (1×10^3 cells/g body weight, median survival 30 h; 1×10^4 cells/g body weight, median survival 45 h) within the same time frame reported by Saville *et al.* for tet-*NRG1* yeast in their severely immunocompromised model. While a direct comparison of the data is difficult due to the inherent differences in the mouse models used, this might support the conclusion that *EED1*-deficient yeast have a higher virulence potential than cells locked in the yeast phase by overexpression of *NRG1*, which would be consistent with observations in immunocompetent animals.

Interestingly, Saville *et al.* also observed a significantly higher fungal burden for tet-*NRG1* yeast compared to tet-*NRG1* filaments in their cyclophosphamide - cortisone acetate model (Table 1 of the paper), although it

NCOMMS-20-24706-T: Point-by-point response

should be noted that the data was not time-matched but acquired from moribund animals. Furthermore, the fungal burden of tet-*NRG1* yeast achieved in this model by Saville *et al.* was similar to (and even slightly higher than) what we observed for *EED1*-deficient yeast in moribund immunocompromised mice (*NRG1*: log 7.94; *EED1*: log 7.23). The fungal burden was also higher than what we observed for tet-*NRG1* yeast in immunocompetent mice 3 days after infection (log 5.5). This indicates, not surprisingly, that host factors – in addition to the intrinsic capacity of cells to utilize nutrients – control the level of fungal proliferation. It also supports the conclusion that high numbers of yeast cells are necessary to induce severe clinical disease.

Finally, the fungal burden reached at humane endpoints in the different immunocompromised mouse models used in Saville *et al.*, 2008 was comparable (with the exception of highly susceptible C5-deficient DBA/2N mice which succumbed at a lower fungal burden), even though the time points of analyses differed. This might indicate that a threshold exists, above which renal fungal burden impacts organ function, and that this threshold is higher for yeast than for hyphae. Data published a while ago by MacCallum and Odds (2005) supports this hypothesis, as they observed comparable kidney burden in mice at the humane endpoint even if animals were challenged with different infectious doses and survived for a different duration.

We have summarized these thoughts in the revised discussion in lines 429-445.

Other issues include:

1. What clinical score threshold did the authors use to define animals as moribund and/or needing to be euthanized?

Response: We thank the reviewer for pointing out that this information was missing. We have modified one sentence in the methods section to clarify that we refer to animals that reached humane endpoints as “moribund” (lines 579-580 in the manuscript with track changes), and added further information to the section on clinical monitoring and scoring: “Mice were humanely sacrificed when they reached the humane endpoints defined as (i) severe lethargy, (ii) hypothermia, or (iii) a cumulative clinical score of ≥ 5 .” (lines 621-624 in the manuscript with track changes).

2. In the phagocytosis and killing assays (described in Fig. S10), did the authors examine the morphology of the fungal cells? The *eed1Δ* mutant was previously reported to initiate hypha formation for several hours, prior to reverting to yeast growth. If this is also true here, this would explain the lack of difference between the wild-type and mutants. It might also suggest that assessing survival after a longer period of incubation would reveal differences, given that hypha formation has been shown to be required for macrophage escape.

Response: We agree with the reviewer that the initial germ tube formation is similar between WT and mutant and that might be the reason why we could not observe any differences when we looked at killing after 2h of co-incubation. We added microscopic pictures of *C. albicans* WT and *eed1Δ/Δ* mutant in the presence and absence of BMDMs after 1 h (time point for the phagocytosis assay), 2 h (time point for the survival assay), and additionally 6 h, 10 h and 24 h as new Supplementary Fig. 12 and Fig. 5c. After 1 and 2 h both strains had formed germ tubes. We repeated the survival analysis after 6 h of co-incubation with murine BMDMs (Fig. 5b). We decided for 6 hours because at that time, macrophages are not yet completely overgrown by the fungus (Supplementary Fig. 12) but the *eed1Δ/Δ* mutant has already switched back to yeast cell growth. These experiments showed that the proportion of *eed1Δ/Δ* mutant cells that was killed by BMDMs was significantly higher than for the WT; however, the absolute numbers of fungal cells determined by CFU plating were significantly higher for the mutant, both in the presence and absence of macrophages (data are included in Fig. 5d). Despite the higher absolute number of *eed1Δ/Δ* mutant cells, the damage of BMDMs induced by the mutant within 24 h of co-incubation was lower than for the WT (Fig. 5e). This new data is presented in the results part of the revised manuscript (lines 168-183 in the manuscript with track changes) and discussed in lines 346-355 (manuscript with track changes).

NCOMMS-20-24706-T: Point-by-point response

3. In describing the phenotype of the t-EED1 strain (lines 75-8), the authors describe it as hyperfilamentous in the absence of DOX. Do they mean it filaments more profusely (higher percentage of cells and/or longer filaments) following hypha induction or do they mean that it filaments inappropriately under yeast growth conditions (= constitutively filamentous)? This could be clarified by the inclusion of some microscopy of the strain grown under yeast and hypha conditions in the presence and absence of DOX (as a panel in one of the supplemental figures).

Response: We apologize for being imprecise in the phenotype description. While filamentation of t-EED1 in the absence of doxycycline in liquid media was comparable to the wildtype (both regarding percentage of filamenting cells and hyphal length; microscopic pictures in new Supplementary Fig. 2b), we observed filamentous colonies on solid YPD under non-hypha-inducing conditions at 25 °C in which the wildtype produced smooth colonies (new Supplementary Fig. 2a). To clarify this we have changed the respective sentence in the manuscript to: "In the presence of doxycycline (t-EED1+) repressed *EED1* expression leads to yeast cell growth. In absence of doxycycline (t-EED1-), the gene is constitutively expressed resulting in increased filamentation on solid media (Supplementary Fig. 2) and filamentation comparable to the WT in liquid media." (lines 82-85 in the manuscript with track changes).

4. How do the authors reconcile the difference in survival rates between THE1-Clp10 +/-DOX groups (Fig. 2c), given that the fungal burden data (presented in Fig. 2e) shows no difference between these two groups?

Response: As stated in the method section, in the first experiments with these strains (survival experiments), mice were treated with doxycycline containing drinking water 3 days prior to infection and throughout the experiments. We noticed that doxycycline treated mice, but not the mice receiving water without doxycycline, lost weight before infection after doxycycline supplementation was started. We also observed lower water consumption in doxycycline-treated cages. Thus, the weight loss was most likely the result of decreased water uptake due to the bitter taste of doxycycline (although sucrose was added to increase acceptance). While no clear clinical signs of dehydration (skin fold test) were observed, reduced fluid uptake likely aggravated the effects of renal impairment induced by infection and thereby accelerated clinical disease. In consequence, we changed antibiotic administration from drinking water to doxycycline containing food for subsequent experiments. The food was accepted well and no differences in weight between the doxycycline treated and untreated groups were observed.

To clarify this point, we have extended the description in the method section (lines 570-572 in the manuscript with track changes) and added a sentence to results part (lines 126-128 in the manuscript with track changes).

An alternative – or additional – explanation could be doxycycline-induced changes in the microbiota with subsequent impact on IL-17 production. This has been recently demonstrated to affect susceptibility of mice to systemic candidiasis by Li *et al.* (Microbiota-driven interleukin-17 production provides immune protection against invasive candidiasis; Critical Care, 2020). In contrast to our study, Li *et al.* however used a combination of five antibiotics not including doxycycline (ampicillin, gentamicin, metronidazole, neomycin sulfate, vancomycin) and treated mice for 2-3 weeks before systemic infection. Thus, it is likely that the impact on the microbiome in the study of Li *et al.* was much more profound than in our experiments where only doxycycline was given for only 3 days before infection. Furthermore, since we did not collect fecal samples for microbiome analysis and did not analyze serum IL-17, we cannot determine whether a similar mechanism affected survival in our study. To avoid unnecessary distraction from the focus of our study, we therefore decided not to discuss the possible impact of antibiotic treatment on microbiota-induced cytokine responses in the revised manuscript.

5. For the fungal burden data reported in supplementary Fig. 2d, mice infected with the t-NRG1 strain in the absence of doxycycline never become moribund so they need to define the specific time when these were presumably humanely sacrificed.

Response: We thank the reviewer for pointing this out. We have added the relevant information to the figure legend: "Mice challenged with tet-*NRG1*- were humanely sacrificed 19 days post infection."

NCOMMS-20-24706-T: Point-by-point response

6. The statement attempting to link the lack of an immune response by epithelial cells infected with the *eed1Δ* mutant to damage (lines 141-4) is inaccurate, at least with respect to the oral TR146 cells. In these, the wild-type strain causes significantly increased damage compared to the mutant, but no increase in cytokine production.

Response: We kindly disagree that the WT does not lead to an increase in cytokine production in oral epithelial cells: While there was indeed no significant increase in IL-8 and IL-6 production observed in TR146 cells infected with the WT compared to the uninfected control, the mean cytokine levels were two-fold higher than in the controls. In contrast, cytokine levels in uninfected samples and samples from cells infected with the deletion mutant were comparable. The lack of statistical significance in these experiments is likely a consequence of the higher variation between the biological replicates, but we did observe a trend. Importantly, oral epithelial cells are not involved in the systemic candidiasis model (since fungi are directly introduced into the blood stream), and we therefore assume that the data for renal and hepatic cells – but not oral cells – might shed light on the *in vivo* infection process in this model. To clarify this issue, we have re-written the respective part of the results to more accurately reflect our findings and conclusion: “Infection of renal, hepatic and oral epithelial cells with the *eed1Δ/Δ* mutant induced no increase of cytokines compared to uninfected controls (Fig. 4b), likely due to the lack of damage caused by the mutant in this model (Fig. 1a). For oral cells, no significant increase was observed for the WT either, although a tendency of higher cytokine production was observed. Thus, the initial lower cytokine response *in vivo* in response to t-EED1+ yeast is likely due to reduced damage of and lower cytokine production by renal and hepatic cells and macrophages, which is balanced by the response of recruited immune cells at later time points.” (lines 185-189 in the manuscript with track changes)

Several minor points:

1. In describing previous virulence studies performed using morphotype locked *C. albicans* strains (line 43), the Braun et al. paper describing the characterization of the *NRG1* gene is cited, but not the companion paper published at the same time by Murad et al. In addition, the word hyphal should be replaced with filamentous, given that the *tup1Δ* mutant is constitutively pseudohyphal.

Response: We agree, and have added a citation for Murad *et al.* and have replaced “hyphal” with “filamentous” (lines 56-58 in the manuscript with track changes).

2. In several figures displaying *in vivo* data (e.g. Figs. 3, 5 and S11), I would suggest referring to the dashed lines as data from control, uninfected mice, rather than simply control (which could refer to the wild-type infections) or untreated (the addition of doxycycline is a treatment) to avoid confusion. Similarly, I presume that the control data presented in supplementary figures 5 through 9 was obtained from uninfected animals.

Response: We agree that “control” can lead to confusion and have thus changed the legend in the graphs or the description in the figure legend to precisely describe the type of control used.

3. The legend for Fig. 1S states that the wild-type strain used in these experiments was BWP17. Given that the *eed1Δ* mutant was constructed in the SC5314 background, and there is no mention of BWP17 in the methods section, I assume that this is an oversight.

Response: We thank the reviewer for pointing this out. We indeed forgot to describe the BWP17 strain and corresponding mutant in the method section. We originally performed the experiment with an *eed1Δ/Δ* mutant in the BWP17 background at the beginning of the project, but recently repeated the experiment with the SC5314 WT strain and the respective *eed1Δ/Δ* mutant with comparable results: Mice infected intraperitoneally

NCOMMS-20-24706-T: Point-by-point response

with the *eed1Δ/Δ* mutant showed less clinical symptoms and less liver and pancreatic damage. To avoid unnecessary confusion we therefore decided to replace the data obtained with the BWP17 background with the new data (SC5314 background) in the supplement (Fig. S1). The figure legend has been changed accordingly.

4. In Fig. 3S, the descriptions for panels a and b are swapped in the legend.

Response: We apologize for the mistake. We have swapped the graphs so that panel “a” now shows the body weight and panel “b” the clinical score, consistent with the figure legend.

5. With regards the multiplex cytokine analysis, the methods section states that a custom 12-plex panel was employed but the data from only 11 are reported in the figures. What happened to the IL-10 data?

Response: We accidentally did not include the data for IL-10 and thank the reviewer for pointing this out. Figure 3b now includes the IL-10 data (no difference between WT and mutant was observed).

6. In several of the figures using the tEED1 strain, the legends refer to the wild-type strain as THE1 when it should THE1-Clp10 (the parent is *ura-* and therefore avirulent in the disseminated model).

Response: The reviewer is right; we indeed used THE1-Clp10. We have corrected this mistake throughout the manuscript.

7. Use of the term human endpoint, rather than humane endpoint in several places.

Response: We have corrected this mistake throughout the manuscript.

8. Line 181, I believe the word should be obverse, rather than adverse in this context.

Response: The reviewer is correct and we have changed the sentence accordingly.

REVIEWERS' COMMENTS

Reviewer #1 (Remarks to the Author):

The authors have carried out a careful and extensive review that has involved further experimentation to address concerns raised by both referees. Although some questions remain to be answered (e.g. the mechanism by which *eed1* depletion affects Y-H transition and growth of alternative carbon sources), the revised version is significantly strengthened and makes some important advances in this field.

Reviewer #2 (Remarks to the Author):

The revisions made in this resubmitted version of the manuscript by Dunker and colleagues are extensive and incredibly responsive to the comments and critiques of the original reviewers. In particular, I commend the authors on performing several additional, not insubstantial experiments, which have significantly strengthened their scientific arguments. The new 6h p.i. BMDM experiments, the enterocyte and GI infection experiments and the lower dose and time matched burden experiments (in immunosuppressed mice) all support the premise that enhanced proliferation of the *eed1* Δ/Δ mutant in kidney tissues is the main determinant of virulence of these yeast form cells. Overall, I believe that the authors have improved the quality of the manuscript enough to warrant publication in Nature Communications. However, there are still a few minor issues that should be addressed prior to publication.

1. To clarify the point, I would strongly recommend rephrasing some of the sentences on page 13 describing the new data from the immunocompromised mouse experiment to read: "Although the *eed1* Δ/Δ mutant led to 100% mortality in immunosuppressed mice, this was significantly delayed in comparison to WT strain infections. However, in immunocompetent mice with ... the experiment (Fig. 3a). In addition, histological analysis of renal tissue ... to yeast driven mortality".
2. Fig.1 panel d still has the uninfected sample listed as ctr (for control), which is inconsistent with the terminology used in panel a (and which the authors agreed to change).
3. Fig.1 legend should indicate "b-e t-EED1+ yeasts are attenuated in invasion ..."
4. Fig. 2 panel a, what is the "high control" used as the comparator (it isn't described in either the legend or the materials and methods sections of the manuscript)?
5. Fig. 2 legend: sampling time-point is missing for panel f.
6. Fig. 7 legend needs a brief description on how the authors are defining "uniqueness".
7. The list of parameters used to determine clinical score differs between the legends of Figs 1 and S3.
8. Figs 5, S6, S7, S8, S9 and S10 still have panels using the "ctr" terminology instead of "uninfected"; for Fig. S12, I suggest -BMDMs, rather than ctr.

Finally, there are quite a few minor grammatical issues in the newly added (highlighted) text, which I am assuming will be identified and corrected in the final editorial process.

NCOMMS-20-24706-T: Point-by-point response

We would like to thank the reviewers for the critical reading and constructive comments that helped us to improve the manuscript. We have addressed all comments and provide a detailed point-to-point reply to all comments below. Changes made in response to the reviewers are highlighted in blue in the revised manuscript.

Reviewer #1 (Remarks to the Author):

The authors have carried out a careful and extensive review that has involved further experimentation to address concerns raised by both referees. Although some questions remain to be answered (e.g. the mechanism by which *eed1* depletion affects Y-H transition and growth of alternative carbon sources), the revised version is significantly strengthened and makes some important advances in this field.

Response: We thank the reviewer for this positive feedback.

Reviewer #2 (Remarks to the Author):

1. To clarify the point, I would strongly recommend rephrasing some of the sentences on page 13 describing the new data from the immunocompromised mouse experiment to read: “Although the *eed1* Δ/Δ mutant led to 100% mortality in immunosuppressed mice, this was significantly delayed in comparison to WT strain infections. However, in immunocompetent mice with ... the experiment (Fig. 3a). In addition, histological analysis of renal tissue ... to yeast driven mortality”.

Response: We thank the reviewer for this constructive suggestion and have changed the sentences accordingly.

2. Fig.1 panel d still has the uninfected sample listed as ctr (for control), which is inconsistent with the terminology used in panel a (and which the authors agreed to change).

Response: We apologize for this oversight and changed “ctr” to “uninfected” in the graph.

3. Fig.1 legend should indicate “b-e t-EED1+ yeasts are attenuated in invasion ...”

Response: We have corrected this error.

4. Fig. 2 panel a, what is the “high control” used as the comparator (it isn’t described in either the legend or the materials and methods sections of the manuscript)?

Response: “high control” refers to supernatant of Triton X-100-lysed uninfected cells (maximum of LDH release). This information has been added to the figure legend and the methods section.

5. Fig. 2 legend: sampling time-point is missing for panel f.

Response: Sampling time-point (14 d p.i.) is now included in legend.

6. Fig. 7 legend needs a brief description on how the authors are defining “uniqueness”.

Response: A description was added: “Color is representing the uniqueness, indicating how particular each term is with respect to the set of terms being evaluated”.

7. The list of parameters used to determine clinical score differs between the legends of Figs 1 and S3.

Response: We thank the reviewer for pointing this out. However, the clinical score is different due to the fact that different infection models were used. Fig 1 shows data from the intraperitoneal infection model, whereas and Fig S3 shows data from the systemic infection model. The two different scoring systems are described in the methods section.

8. Figs 5, S6, S7, S8, S9 and S10 still have panels using the “ctr” terminology instead of “uninfected”; for Fig. S12, I suggest –BMDMs, rather than ctr.

Response: We apologize for this oversight and have changed ctr to uninfected in Fig.5 and figures S6-S10. In Fig. S12 “ctr” was changed to “without BMDMs”.

Finally, there are quite a few minor grammatical issues in the newly added (highlighted) text, which I am assuming will be identified and corrected in the final editorial process.

We have carefully checked the text and corrected any mistakes we identified.